# Australian Lidar Measurements of Aerosol Layers Associated with the 2015 Calbuco Eruption

**Andrew R. Klekociuk** [1,2], **David J. Ottaway** [2], **Andrew D. MacKinnon** [2], **Iain M. Reid** [2,3,*],
**Liam V. Twigger** [2] **and Simon P. Alexander** [1]

[1] Department of the Environment and Energy, Australian Antarctic Division, Antarctica and the Global System, Kingston, TAS 7050, Australia; andrew.klekociuk@aad.gov.au (A.R.K.); simon.alexander@aad.gov.au (S.P.A.)

[2] School of Chemistry and Physics, University of Adelaide, Adelaide, SA 5005, Australia; david.ottaway@adelaide.edu.au (D.J.O.); andrew.mackinnon@adelaide.edu.au (A.D.M.); liam.twigger@protonmail.com (L.V.T.)

[3] ATRAD Pty. Ltd., 20 Phillips St., Thebarton, SA 5031, Australia

\* Correspondence: iain.reid@adelaide.edu.au

**Abstract:** The Calbuco volcano in southern Chile (41.3° S, 72.6° W) underwent three separate eruptions on 22–23 April 2015. Following the eruptions, distinct layers of enhanced lidar backscatter at 532 nm were observed in the lower stratosphere above Buckland Park, South Australia (34.6° S, 138.5° E), and Kingston, Tasmania (43.0° S, 147.3° E), during a small set of observations in April–May 2015. Using atmospheric trajectory modelling and measurements from the Cloud-Aerosol Lidar with Orthogonal Polarization (CALIOP) space-borne lidar and the Ozone Mapping Profiler Suite (OMPS) instrument on the Suomi National Polar-orbiting Partnership (NPP) satellite, we show that these layers were associated with the Calbuco eruptions. Buckland Park measurements on 30 April and 3 May detected discrete aerosol layers at and slightly above the tropopause, where the relative humidity was well below saturation. Stratospheric aerosol layers likely associated with the eruptions were observed at Kingston on 17 and 22 May in narrow discrete layers accompanied by weaker and more vertically extended backscatter. The measurements on 22 May provided a mean value of the particle linear depolarisation ratio within the main observed volcanic aerosol layer of 18.0 ± 3.0%, which was consistent with contemporaneous CALIOP measurements. The depolarisation measurements indicated that this layer consisted of a filament dominated by ash backscatter residing above a main region having likely more sulfate backscatter. Layer-average optical depths were estimated from the measurements. The mean lidar ratio for the volcanic aerosols on 22 May of 86 ± 37 sr is consistent with but generally higher than the mean for ground-based measurements for other volcanic events. The inferred optical depth for the main volcanic layer on 17 May was consistent with a value obtained from OMPS measurements, but a large difference on 22 May likely reflected the spatial inhomogeneity of the volcanic plume. Short-lived enhancements of backscatter near the tropopause of 17 May likely represented the formation cirrus that was aided by the presence of associated volcanic aerosols. We also provide evidence that gravity waves potentially influenced the layers, particularly in regard to the vertical motion observed in the strong layer on 22 May. Overall, these observations provide additional information on the dispersal and characteristics of the Calbuco aerosol plumes at higher southern latitudes than previously reported for ground-based lidar measurements.

**Keywords:** stratosphere; volcanic aerosol; lidar

## 1. Introduction

Stratovolcanoes are characterized by periodic and violent eruptions, which are capable of forcing large fluxes of silicate-bearing ash, sulfate-containing compounds (primarily sulfur dioxide ($SO_2$)

and carbonyl sulfide (OCS)) and other volatiles into the stratosphere. When in the stratosphere, $SO_2$ oxidizes into sulfuric acid vapor, and is then converted to sulfuric acid particles by homogeneous nucleation. This conversion process operates over timescales of several weeks. A secondary source of stratospheric $SO_2$ is produced by oxidation of OCS in the stratosphere [1]. Solid particles, which may grow through condensation and coagulation, undergo gravitationally sedimentation but may also be lofted by the residual meridional circulation in the rising branch of the Brewer–Dobson circulation and more local small-scale motions [2]. Typically, stratospheric residence of particles range from several months to years for sub-micron size particles, and hours to a few days for larger particles [3]. The effects of stratospheric volcanic aerosols include a direct negative forcing on the radiation balance through scattering and absorption of short-wave radiation, and indirect climate effects on cloud nucleation and heterogeneous chemical reactions [1].

On 22 April 2015, the Calbuco stratovolcano in southern Chile (41.3° S, 72.6° W) underwent its first major eruption since 1961 [4]. The eruptive activity of the volcano was relatively brief and explosive, occurring in two main events at 22 April 18:11 Universal Time (UT) and 23 April 04:00 UT, followed by a third weaker eruption on 30 April [5,6]. The first eruptive event cast an aerosol plume as high as 16 km above sea level in a period of several minutes, while the second event was more violent and produced a plume that reached a height of approximately 23 km [7]. The plume from the third event was confined to below 4.5 km altitude [6].

Bègue et al. [8] observed the fine mode aerosol from Calbuco in the altitude range 18–21 km over Reunion Island (21.0° S, 55.5° E) during May to July 2015. They found that the aerosol increased the stratospheric aerosol optical depth by a factor of approximately 2 over background levels. In addition, the e-folding time of the decaying stratospheric aerosol mass was found to be approximately 90 days. Carn et al. [9] estimated the flux of $SO_2$ into the upper-troposphere lower-stratosphere region from the eruption as 0.2 to 0.5 Tg, which placed the event as one of the largest in the Southern Hemisphere over the previous decade. The effects of the Calbuco aerosol plume were not confined to mid-latitudes and included the enhancement of ozone destruction over Antarctica during the austral spring of 2015 [10–13]. The characteristics of the Calbuco aerosol plume and its movement to southern high latitudes were described by Stone et al. [12] from measurements by the Cloud-Aerosol Lidar with Orthogonal Polarization (CALIOP) lidar on the Cloud-Aerosol Lidar and Infrared pathfinder Satellite (CALIPSO) satellite.

In the 10 year period prior to the Calbuco eruption, there were two volcanic events that resulted in appreciable impacts on the upper troposphere/lower stratosphere of the extratropical Southern Hemisphere; Puyehue-Cordon (at 41° S in Chile, 4 June 2011) and Copahue (at 38° S in Argentina, 22 December 2012) [1,9]. Lidar analysis of aerosols from the Puyehue-Cordon eruption were provided by Nakamae et al. [14] who found evidence of non-spherical particles in the high depolarisation ratios that were observed up to a month after the eruption. Vernier et al. [15] observed color ratios between 1064 and 532 nm CALIOP measurements of 0.5 in ash layers 3 weeks after the eruption indicating persistence of irregular particles. Bignami et al. [16] using measurements by the Moderate Resolution Imaging Spectroradiometer (MODIS) satellite instrument found that the mean particle radius in the plume remained near 4–5 μm during the 4 week period following the eruption. The amount of $SO_2$ lofted into the upper-troposphere lower-stratosphere region from the December 2012 Copahue eruption was estimated as 0.2–0.5 Tg, the same as for Puyehue-Cordon [9], although most of the plume was confined to the troposphere (see Figures 4 and 7 of Carn et al. [9]).

There is a clear hemispheric bias in volcanic emissions, with most large events since the late 1970s being confined to the tropics and Northern Hemisphere extratropics (see Figure 1 of Carn et al. [9]). Ground-based lidar instruments offer advantages to satellites in obtaining details of volcanic aerosols, mainly through their ability to achieve high signal-to-noise ratio and to continuously monitor changing conditions. However, the network of lidar systems capable of measuring into the stratosphere in the Southern Hemisphere is still relatively sparse. As highlighted by Zhu et al. [13], gaps remain in details of stratospheric processes involving volcanic aerosols, particularly in regard to their interaction with

ozone chemistry, and transport across dynamical barriers such as the tropopause and the Southern Hemisphere stratospheric polar vortex.

Here, we describe a small set of lidar measurements of the Calbuco volcanic aerosols made at Buckland Park, South Australia, and Kingston, Tasmania, during April–May 2015. These measurements add to information on the characteristics of volcanic aerosols at middle southern latitudes. In Section 2, we describe our equipment and methods. This is followed in Section 3 by presentation of our results, where we demonstrate the connection between observed layers of enhanced backscatter and expected locations of the volcanic plume, and provide retrievals of particle linear depolarisation ratio, optical depth and lidar ratio. In Section 4, we examine radiosonde and related data at the heights of the observed layers and compare our observations with other similar volcanic measurements, before providing conclusions in Section 5.

## 2. Experiments

### 2.1. Buckland Park Rayleigh/Mie/Raman Lidar

The important parameters of the Buckland Park Rayleigh/Mie/Raman Lidar (situated near Adelaide, South Australia) are provided in Table 1. More details are provided in Reid et al. [17]. Briefly, the main components of the lidar are a zenith-pointing telescope with a 1 m diameter spherical primary mirror of 8 m radius of curvature and a high power Nd:YAG pulsed laser. The primary mirror is corrected for spherical aberration by a commercial lens placed just short of the focal plane. Pulsed 532 nm light from the laser is expanded to reduce divergence and then directed by a series of steering mirrors so that it is transmitted coaxially with the optical axis of the telescope. Received light at the focal plane of the primary mirror is relayed by a multimode fiber with a 1.8 mm diameter to an optical bench in a light-proof detection room.

**Table 1.** Main parameters for the Buckland Park and Kingston lidars.

| Parameter | Buckland Park | Kingston |
|---|---|---|
| Location | 34.6° S, 138.5° E (~40 km NNW of Adelaide, South Australia) | 43.0° S, 147.3° E (~12 km S of Hobart, Tasmania) |
| Operating mode | Simultaneous Rayleigh/Mie (532 nm) and $N_2$ Raman (608 nm) backscatter | Rayleigh/Mie (532 nm) backscatter; time-multiplexed co- and cross-polarised measurements were available for some periods |
| Laser energy per pulse and wavelength | 400 mJ, 532 nm | 75 mJ, 532 nm |
| Interference filter width (FWHM) | 0.3 nm (532 nm), 1.5 nm (608 nm, blocking >$10^6$ at 532 nm) | 0.1 nm |
| Laser repetition rate | 50 Hz | 50 Hz |
| Dwell per accumulation | 40 s (2000 shots) | 3 s (150 shots)—this was also the switching interval for depolarisation measurements |
| Range resolution | 50 m | 75 m |
| Receiver aperture | 1000 mm | 280 mm |
| Receiver configuration | coaxial full overlap above ~20 km range | bistatic, separation 160 mm full overlap above 900 m range |
| Receiver field of view (full angle) | 0.3 mrad | 0.5 mrad |
| Divergence of transmitter (full angle) | 0.1 mrad | 0.1 mrad |

The light output from the relay fiber is collimated and then separated by a dichroic beamsplitter into separate detection paths for 608 nm molecular nitrogen ($N_2$) rotation-vibration Raman backscatter and 532 nm Rayleigh/Mie backscatter. Light in the Raman path is detected by photomultiplier tube (PMT) operated at room temperature. Light in the Rayleigh/Mie path is split into low- and high-sensitivity channels by a 10/90 beamsplitter. The low-sensitivity channel is detected by a room temperature PMT and is used for measurements in the troposphere and lower stratosphere (1–35 km altitude, including Mie backscatter). The high-sensitivity channel is detected by a cooled PMT and is used for sensing the upper stratosphere and mesosphere (30–90 km, mainly from Rayleigh backscatter). The PMT pulses are collected with a Licel photon-counting system. A high-speed rotating shutter (mechanical chopper) with selectable and stable phase delay relative to the firing of the laser can be introduced to attenuate the bright backscatter from low altitudes; this was fully open for the measurements presented here.

The system is optimized for stratospheric and mesospheric studies. In the troposphere and lower stratosphere, the system has a range-dependent geometric overlap function because the focal waist of the telescope is larger than the diameter of the optical fiber for these ranges. For the measurements considered here, the overlap function affected the signal profile below ~20 km. The approach used to correct for the overlap function is discussed in Section 2.3.

## 2.2. Kingston Depolarisation Lidar

The characteristics of the 532 nm depolarisation lidar at Kingston, Tasmania, are described in Huang et al. [18] and summarized in Table 1. The system is of relatively low power and optimised for tropospheric cloud measurements. During the observations used here, the lidar was being further developed and tested for a forthcoming field campaign, and the system was mainly configured to measure total backscatter using a single channel receiver. Two sets of observations were made using a time-multiplexed depolarisation analyser as used in Huang et al. [18]. In this set up, co-polarised and cross-polarised measurements were alternately made for intervals of 150 laser shots (three seconds) using a liquid-crystal variable retarder (LCVR) followed by a polarization analyser with a 500:1 contrast ratio. The LCVR and analyser, both manufactured by Meadowlark Optics, were the same as used in Seldomridge et al. [19]. The signal was detected with a single cooled PMT and recorded using photon counting with a FASTComTec multichannel scaler card. A high-speed rotating shutter was used to block bright backscatter below 4 km for the observation on 17 May.

Information on the specific observing sessions conducted at Kingston and Buckland Park is presented in Table 2.

**Table 2.** Tropospheric and stratospheric observations from the Buckland Park and Kingston lidars in April–May 2015 analysed at 100 m vertical resolution.

| Date (2015) | Site | Observation Time Span (hour:minute (HH:MM) Universal Time (UT)) | CSR (km) | Time (HH:MM UT) of Radiosonde (Peak Height, km), AIRS Pass (Distance, km) [1] | Identified Aerosol Layers | |
|---|---|---|---|---|---|---|
| | | | | | Altitude Range (km) | Peak 532 nm Scattering Ratio [2] (Height (km), θ (K) [3]) |
| 30 April | Kingston | 30 April 02:40–1 May 06:53 (depolarisation throughout) | (20,25) | 30 April 11:15 (20.6), 30 April 14:53 (23) | Nil [4] | <1.19 (0.07) over 10–20 km |
| | Buckland Park | 12:00–13:00 | (30,35) | 11:16 (24.3), 16:29 (42) | 10.9–12.9 | 1.5 ± 0.1 (11.3, 336) 1.5 ± 0.1 (12.1, 349) 1.2 ± 0.1 (12.7, 356) |
| 3 May | Buckland Park | 12:00–13:41 | (30,35) | 11:17 (24.5), 15:27 (34) | 15.3–16.2 | 1.5 ± 0.1 (15.7, 378) |
| 11 May | Kingston | 09:53–10:55 | (25,30) | 11:14 (21.6), 14:40 (37) | Nil [3,4] | <1.4 (0.2) over 10–20 km |
| 17 May | Kingston | 07:55–09:30 | (25,30) | 11:14 (23.7), 04:34 (26) | 11.0–15.2 17.2 | 1.2 ± 0.1 (11.4, 330) 1.4 ± 0.1 (14.5, 382) 1.1 ± 0.1 (17.2, 433) [6] |
| 22 May | Kingston | 07:24–08:26 (depolarisation 07:52–08:09) | (25,30) | 11:15 (22.9), 04:52 (29) | 14.0–17.0 17.2–18.5 18.7–19.1 | 1.4 ± 0.1 (15.6, 408) 5.2 ± 0.4 (17.8, 446) 3.2 ± 0.3 (19.0, 470) |
| 28 May | Kingston | 21:54–22:28 | (20,25) | 23:16 (29.1), 15:17 (29) | Nil [4,5] | <1.6 (0.4) over 10–20 km |

[1] AIRS is the Atmospheric Infrared Sounder satellite instrument. [2] Uncertainties are 1 σ (1 standard deviation). Single values in parentheses () are the 1σ uncertainty in the upper limit. [3] θ is the potential temperature at the layer peak inferred from radiosonde measurements. [4] Nil means that no discrete layers (local maxima) were observed that were significantly different to the 10–20 km mean scattering ratio at the ± 1σ level. [5] Analysed at 200 m vertical resolution. [6] Analysed at 500 m vertical resolution.

## 2.3. Scattering Ratio Profiles

The signal P(z) as a function of vertical height z from our two lidars can be represented by

$$P(z) = T_s(z) \times (\{C \times (\beta_{mol}(z;\lambda) + \beta_{aer} \times (z;\lambda)) \times T_{mol}(z;\lambda)^2 \times T_{ozone}(z;\lambda)^2 \times T_{aer}(z;\lambda)^2\} \div z^2) \quad (1)$$

where $T_s$ is a height-dependent transmission term representing the combined effect of the geometrical overlap function and the rotating shutter used in our systems, C is a calibration constant representing instrumental quantities, $\beta_{mol}$ and $\beta_{aer}$ are the molecular and aerosol backscatter coefficients, respectively, and $T_{mol}$, $T_{ozone}$ and $T_{aer}$ are optical transmission factors of air molecules, ozone and aerosols, respectively, from the lidar to height z, and λ indicates a wavelength dependence. For the Kingston measurements, we set $T_s(z) = 1$ (as the effects of the overlap function and any shutter attenuation were confined to the lower troposphere). For the Buckland Park measurements, $T_s$ was less than 1 below approximately 20 km and influenced our retrievals as discussed below. For Equation (1) we also have

$$\beta_{mol}(z) = \sigma_{air}(\lambda) \times N_{air}(z) \quad (2)$$

where $\sigma_{air}$ and $N_{air}$ are the Rayleigh backscatter cross section and molecular number density for air, respectively. The transmission terms in Equation (1) are given by

$$T_x(z;\lambda) = \exp\{- \int_{z_0}^z \alpha_x(r;\lambda) \times dr\} \quad (3)$$

where *x* denotes the relevant species (air molecules, ozone molecules or aerosols), $z_0$ is the height of the lidar above mean sea level and $\alpha_x$ is the extinction coefficient of species *x*.

To evaluate the transmission term for air as well as the molecular backscatter coefficient $\beta_{mol}$, we obtain the molecular number density profile at the same height bins as our lidar measurements from measurements of temperature as a function of pressure obtained using a blend of contemporaneous radiosonde and satellite data. Radiosondes launched near our sites provided density profiles at 5 m

vertical resolution to heights of at least 20 km, above which we concatenated a molecular profile derived from the closest available Atmospheric Infrared Sounder (AIRS) version 6 level 2 data [20] The typical difference between the radiosonde and AIRS molecular densities at the top of each radiosonde profile (typically 21–24 km) was ~2%. The necessary ozone number density profile for the ozone transmission term was also obtained from the AIRS data product. Not including the ozonecorrection increases the retrieved scattering ratios by up to ~0.04 in the troposphere, and this bias decreases towards zero at the top of the ozone layer (~30 km).

For the Buckland Park Raman channel, where the transmitted and received wavelengths were 532 and 608 nm.

$$T_x(z;\lambda)^2 = T_x(z;\lambda_{532}) \times T_x(z;\lambda_{608}) \tag{4}$$

where the subscripts for $\lambda$ denote the relevant wavelength. The backscatter and extinction coefficients are related by the lidar ratio S:

$$S_{aer}(z;\lambda) = \alpha_{aer}(z;\lambda) \div \beta_{aer}(z;\lambda) \tag{5}$$

$$S_{mol}(z;\lambda) = \alpha_{mol}(z;\lambda) \div \beta_{mol}(z;\lambda) = 8\pi/3 \tag{6}$$

Here, we assume single scattering, and that the mean molecular mass is constant with altitude. Also, implicit in Equation (1) is that the background due to detector noise and background light has been removed. The sky background contribution was in any case negligible compared with the backscattered signal as the measurements were conducted at night during a new moon period.

For our analysis we obtained the scattering ratio R(z) which is defined as

$$R(z) = (\beta_{aer}(z) + \beta_{mol}(z)) \div \beta_{mol}(z). \tag{7}$$

For the Kingston measurements, Equation (1) was inverted to obtain profiles of $\alpha_{aer}$ constrained by $\beta_{mol}$ and $S_{aer}$ using downward iteration of the standard Klett–Fernald–Sasano method, which uses the approach of Klett [21] and Fernald [22] with allowance for height-varying aerosol lidar ratio as provided in Sasano et al. [23]. Using Equation (2) we calculated $\beta_{aer}$ from $\alpha_{aer}$ and the assumed profile of $S_{aer}$, and then obtained R by Equation (3).

For Buckland Park, the retrieval was not ideally constrained due to the influence of the geometric overlap function for which we do not have an accurate a priori characterization and unfortunately the standard Raman retrieval method of Ackermann et al. [24] could not be applied. To provide a constraint to our inversion, we first assumed that the aerosol extinction at the two wavelengths (532 and 608 nm for Rayleigh and Raman backscatter, respectively) was related by a particular value of the Ångström exponent Å. Here, Å is defined such that

$$\tau_1/\tau_2 = (\lambda_1 \div \lambda_2)^{-\text{Å}} \tag{8}$$

where $\tau$ is the optical depth at wavelength $\lambda$ [25], with

$$\tau = -\ln T \tag{9}$$

where T is the transmission fraction of the aerosol. We specifically assume the aerosols over our retrieval heights are characterized by Å = 1. This value is consistent with sun photometer measurements of volcanic aerosols from the 2010 Eyjafjallajökull volcanic eruption. Mortier et al. [26] observed values between ~0.65 and ~1.9 in the ~1 month period following the eruption. For the same Icelandic eruption, Kokkalis et al. [27] found that the range-resolved Ångström exponent obtained between 355 nm and 532 nm ranged from 0.7 to 1.7 and varied as the volcanic layers travelled to the Eastern Mediterranean. Background stratospheric aerosols, on the other hand, show values of $\alpha$ between ~2.4 and ~3.4 [28], but we assume that the low extinction of these aerosols did not have a significant effect on our retrieval.

The first step in the retrieval was the determination of the calibration constant C in Equation (1) at each wavelength. Assuming that there were no aerosols between heights $z_1$ and $z_2$, termed the Clear

Signal Region (CSR), we rearranged Equation (1) using the calculated molecular and ozone transmission terms to obtain the calibration constant C at each wavelength from a weighted least-squares over the CSR using the profile of molecular density. For our purposes we used a CSR of 30–35 km for the Rayleigh channel (typically above significant stratospheric aerosols), and 20–35 km for the Raman channel (lower than for the Rayleigh measurements due to the lower sensitivity of this channel, but over heights where aerosol effects were still expected to be negligible). The lower height of both CSRs was at or above the lowest height where there was full overlap between the transmitter and receiver. Assuming an Ångström exponent of 1 and that $\beta_{aer}(z) = 0$ for the Raman channel, we then solved for $T_s$ and then applied the Klett–Fernald–Sasano approach. In performing this analysis, we set $T_s = 1$ above 20 km. This choice is justified on the basis that a noticeable reduction in R below 1 occurred below this height when applying the Klett–Fernald–Sasano retrieval assuming no aerosol backscatter or extinction. A further assumption is that $T_s$ was not significantly wavelength dependent, which is reasonable considering that the main collecting optic in the receiver is reflective.

To initialize both the Buckland Park and Kingston retrievals, we provide as an upper boundary condition to the Klett–Fernald–Sasano method an estimate of the molecular backscatter coefficient $\beta_{mol}(z_{ref})$ at reference height $z_{ref}$. This was done by using the calibration constant C from a fit of Equation (1) over the CSR (assuming no aerosol influences), and then Equation (1) was rearranged to calculate $\beta_{mol}$ at height $z_2$ (e.g., as done in [29]). We found that this method gave robust retrievals of R that rapidly converged towards the base of the CSR in the face of photon noise and that did not strongly depend on the starting height. For example, when the retrievals were started at different heights down to 5 km below $z_{ref}$, the difference between the obtained scattering ratio compared with that for the retrieval started at $z_{ref}$ was less than 0.001.

## 2.4. Depolarisation Measurements

The volume linear depolarisation ratio obtained with the Kingston lidar was obtained from

$$\delta_v(z) = \beta\perp(z) \div \beta\|(z) \tag{10}$$

where $\beta\perp(z)$ and $\beta\|(z)$ are the volume backscatter coefficients (which include molecular and aerosol components) for cross-polarised and co-polarised states, respectively, as a function of height. Here, we assume that the measured depolarisation does not change significantly over the six-second full duty cycle of the LCVR.

The potential main calibration-related issues for the depolarisation measurements with the Kingston system are (a) PMT non-linearity in the face of high signal levels (potentially affecting the co-polarised measurements more than the cross-polarised measurements because of the greater level of the former), (b) the LCVR states not being orthogonal due to incorrect voltage setting when switching between states, (c) there being an offset angle between the planes of polarization of the outgoing laser light and the co-polarization plane of the analyser, and (d) depolarisation by non-reflective optics and thin-film coatings. Only one detector and common optics were used throughout the system and any optical transmission effects were not significant for the two polarization states. For (a), we operated the PMT with maximum effective count rates below ~1 MHz to minimize possible non-linearity to < 2% (e.g., [30]). For (b), the voltages required to switch the LCVR between the two orthogonal states were obtained from calibration information provided with the device and verified in the laboratory using a calibration laser fitted with a Glan-Taylor polarizer. For (c), the orientation of the axes of the polarization analyser relative to the horizontal plane of polarization of the laser was optimized using aerosol-free night-time backscatter in the mid-stratosphere (above 25 km) to minimize (maximize) the observed signal in the cross-polarised (co-polarised) state (e.g., as suggested in [31]). This was verified by manually introducing a Lyot depolarizer before the polarization analyser and obtaining equal signals, within photon counting uncertainties, in both states. No analytical correction of the polarization phase angle (e.g., through Equation (4) of [32]) was applied in the analysis presented here.

For (d), we had only 2 uncoated transmitting optics before the polarization analyser. As we note in Section 3.4, we estimate from photon-counting statistics of the calibration in (c) that any systematic bias was less than 5% at the 1σ (1 standard deviation) significance level.

## 3. Results

### 3.1. Buckland Park Scattering Ratio

Figure 1 summarizes retrieved scattering ratio profiles from session-average measurements at Buckland Park using the analysis method described in Section 2.3. Initially, we used $S_{aer}(z) = 50$ sr typical of background stratospheric aerosol [33], and then used a value of 65 sr, typical of volcanic aerosol [34] over the layers identified likely being associated with the eruption from the trajectory analysis discussed below. A difficulty with the Buckland Park retrieval was the level of photon noise in the Raman channel which influenced the correction for the overlap function. In panels (a) and (c) of Figure 1 we show the full retrieval, and in panels (b) and (d) we have smoothed $T_s$ with a 10th order polynomial (fitted over 2–20 km) and then retrieved the scattering ratio using the Rayleigh channel signal. This approach removed some apparently spurious spikes in the upper levels, such as near 18 km for the 30 April observation which was due to detector noise in the Raman channel. Overall, the Buckland Park measurements provide reasonable quantitative information on the location and backscatter coefficient of aerosol layers but are not sufficiently robust to evaluate extinction.

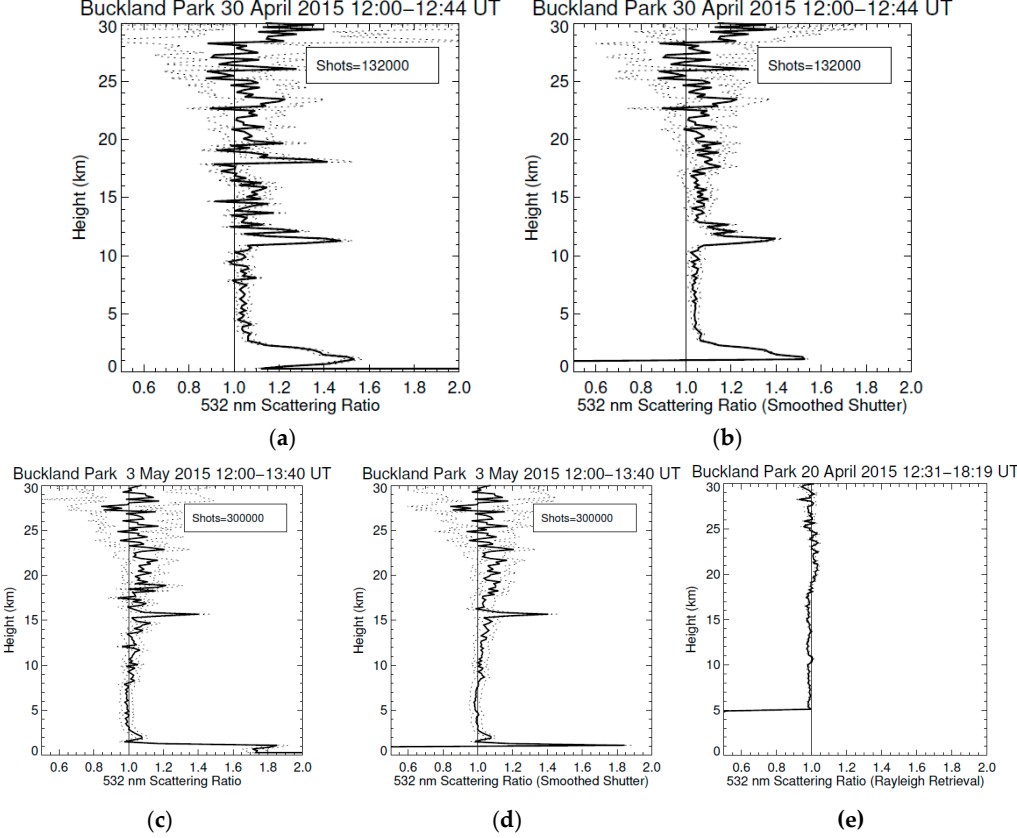

**Figure 1.** Retrieved 532 nm mean scattering ratio for Buckland Park lidar measurements for (**a**) and (**b**) 30 April and (**c**) and (**d**) 3 May 2015 using 200 m vertical resolution. In panels (**b**) and (**d**), the correction for the transmission of the rotating shutter has been smoothed with a polynomial fit as discussed in Section 3.1. The solid vertical line marks unit scattering ratio. Thin dashed lines mark 1σ (1 standard deviation) significance limits. (**e**) Scatting ratio for measurements on 20 April 2015 before the Calbuco eruption. The vertical resolution is 200 m. The profile from 5 to 20 km has been adjusted for height-dependent attenuation the geometric overlap function discussed in Section 3.1.

As can be seen in Figure 1a–d, distinct layers of enhanced scattering ratio were apparent near 12 km height on 30 April and near 16 km height on 3 May, and these features persisted throughout each observing session. The scattering ratio analyses for the Buckland Park and Kingston observations are summarized in Table 2. The CSR used for normalization of the profiles is listed in Table 2, along with details of the associated radiosonde and AIRS measurements. In Figure 1e, we show the scattering ratio profile obtained from Buckland Park measurements on 20 April, before the eruption of Calbuco. We did not make $N_2$ Raman measurements on this occasion. The correction for the overlap function in this case was performed by applying the smoothed version of $T_s$ obtained from the 30 April observation. The profile in Figure 1e is devoid of strong scattering layers, and thus likely represented the background conditions.

We also note that troposphere appeared generally free of aerosol backscatter in the Buckland Park observations, with the scattering ratio being generally less than 1.1 over the height range 5–10 km (Figure 1b,d,e). This was also the case for Kingston (Section 3.2 below). Similar low scattering ratios in the free troposphere are apparent in the long-term measurements from Lauder, New Zealand (45° S) reported by Nagai et al. [35]. It is possible that the scattering ratio retrievals at the lowermost altitudes our lidar measurements were influenced by pulse pile-up causing underestimation of the true count rate [30]. However, any significant effects were likely to be confined below ~7 km where the effective count rates in all channels of our systems were greater than ~100 kHz.

For the Buckland Park observations on 30 April and 3 May, we examined contemporaneous version v4.10 browse data from the CALIOP lidar on the CALIPSO satellite [36]. Figure 2 shows selected browse data showing enhanced backscatter near to the heights of the layers observed at Buckland Park. The CALIOP measurements were spatially and temporally closest to the Buckland Park measurements as follows: 30 April (time separation +4.1 h, distance 615 km west) at 33.7° S, 131.9° E; 3 May (time separation +2.6 h, distance 893 km east) at 36.8° S, 148.0° E. The identified CALIOP backscatter feature on 30 April was centered approximately 935 km south-west of Buckland Park, and above the tropopause at a height of 13 km, consistent with the upper part of the scattering layer shown in Figure 1b. Although the enhanced backscatter feature is reasonably distinct in the CALIOP browse data, it was not identified as a cloud or aerosol feature by the v4.10 analysis algorithm. At the location of the minimum distance from Buckland Park (left edge of Figure 3a), a cirrus cloud layer near 10 km was classified as containing the 'dust' (yellow; index 2) aerosol subtype. For the layer on 3 May, this was identified as 'sulfate/other' in the CALIOP analysis.

To examine the association of the features in Figures 1 and 2 discussed above, we applied trajectory analysis from the Hybrid Single Particle Lagrangian Integrated Trajectory Model (HYSPLIT; [37]) version svn:854. Our analysis used meteorological reanalysis from the European Centre for Medium-Range Weather Forecasts Interim Reanalysis (ERA-Interim; [38]) at the native resolution of 0.75° in latitude and longitude. ERA-Interim provides a good representation of near-surface winds in the Southern Hemisphere in comparison with other leading reanalyses [39]. For the forward and backward trajectory runs shown in Figure 3a,b, respectively, there is reasonable consistency that the enhanced backscatter features observed with the Buckland Park lidar between 11 and 13 km height on 30 April link back to the Calbuco eruption on 22 April. In addition, the CALIOP backscatter feature shown in Figure 3a also appears consistent with the arrival location of trajectories linking back to the 22 April eruption. Similar consistency was found using trajectories run with Global Forecast System final analysis data at 0.5° horizontal resolution (not shown) [40,41]. For the layer at 15–16 km height on 3 May shown in Figures 1c and 2b, a similar connection with the region of Calbuco at the general time of the 22–23 April eruptions was also found. This is shown in the forward and backward trajectories maps in Figure 4a,b, respectively. In Figure 4a, forward trajectories originating at the location of Calbuco at the time of the second major eruption appeared to arrive near Buckland Park on 3 May, at least for trajectories analysed as being near 14–15 km height. A similar connection was apparent if the starting time of the first eruption was used (not shown), although a less clear connection for 15–16 km height was apparent. This could be due to the length of the trajectory runs. Normally 144 h (6 days) is a

suggested upper limit for useful analysis [42] and the end times for the trajectories shown in Figure 4a, and also Figure 3 for that matter, are well beyond this limit. However, we suggest that the length of our performed runs appears to be suitable for source-observation comparisons, because as shown in Figure 3b, the trajectories that are consistent with the height of the CALIOP aerosol layer in Figure 3b have a connection to the region of the Calbuco volcano on April 23 (green and blue trajectories in Figure 3b). Of note in the analysis shown in Figure 4 is the spreading of the trajectories over southern Africa on or approximately on 29–30 April. This appears due to an upper level longwave feature in the region [43] which may have been associated with a tropopause fold.

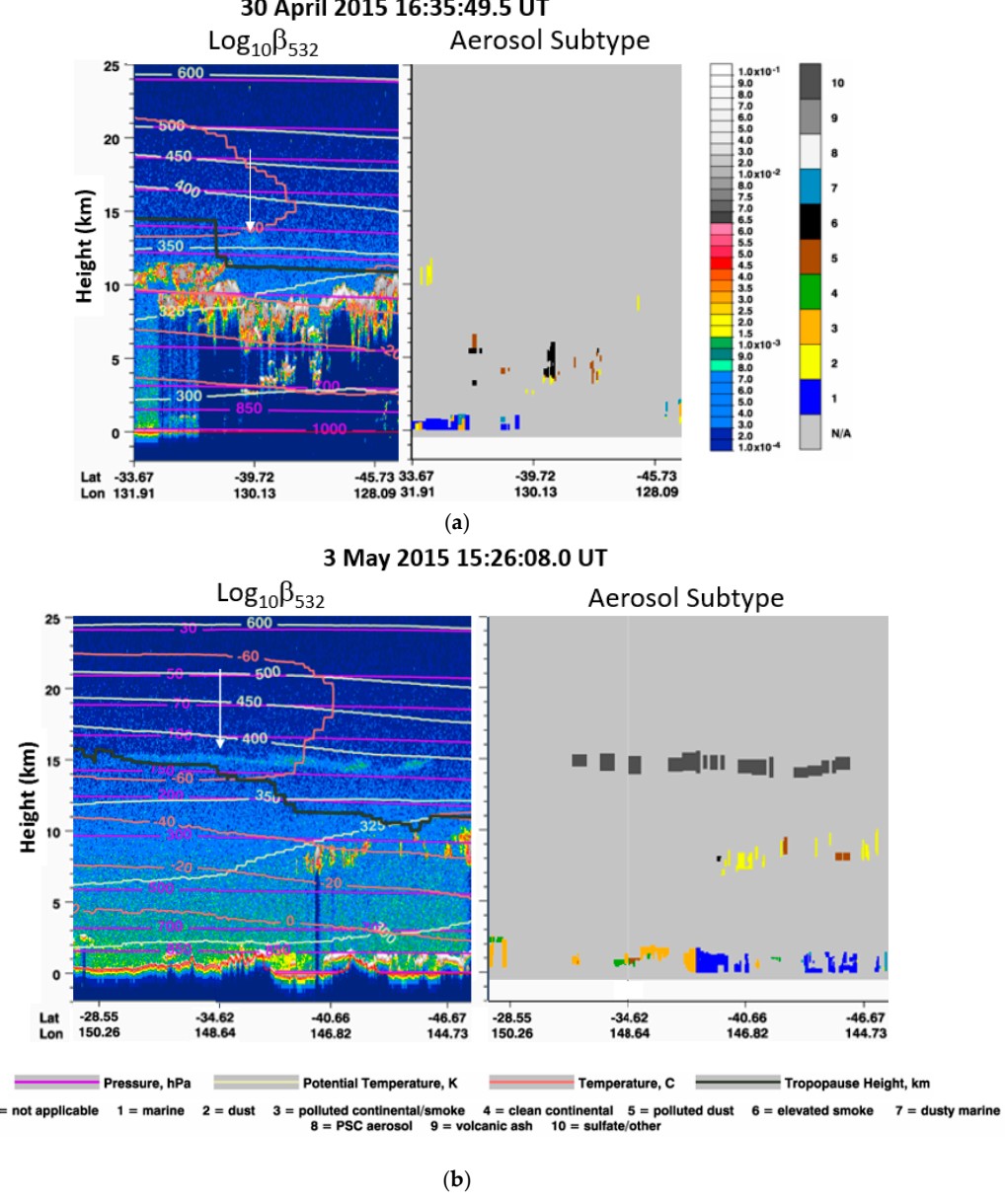

**Figure 2.** Selected Cloud-Aerosol Lidar with Orthogonal Polarization (CALIOP) analysis from version 4.10 browse images for (**a**) 30 April and (**b**) 3 May 2015. Shown for each date are height-time displays of unattenuated backscatter at left and analysed aerosol subtype at right. The color scales for backscatter and aerosol subtype are shown at right of panel (**a**). The legends for the contours in the backscatter panels and the filled regions in the aerosol subtype panels are shown at the bottom. The time in the heading indicates the starting time of each measurement curtain. The vertical white arrows highlight regions of enhanced stratospheric backscatter discussed in Section 3.1.

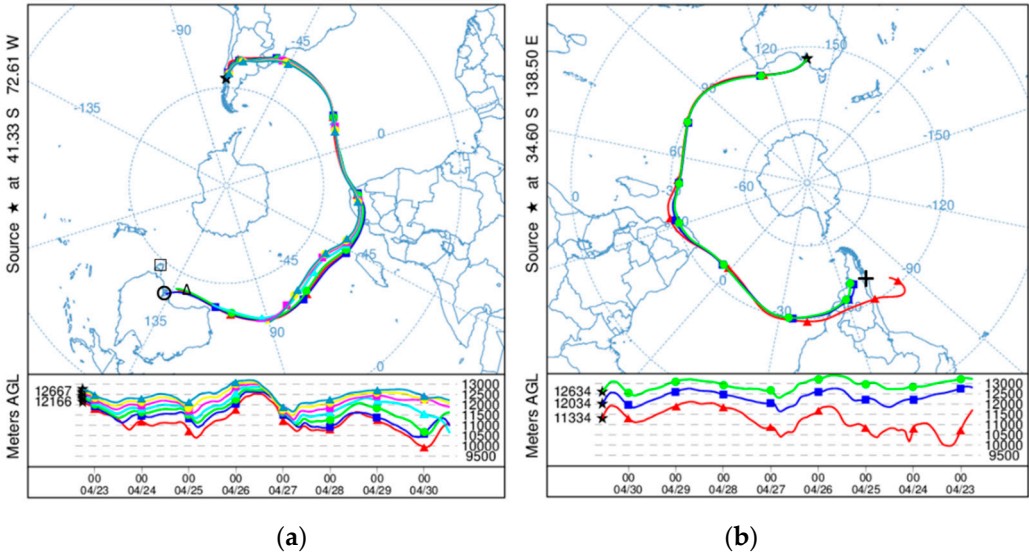

**Figure 3.** Hybrid Single Particle Lagrangian Integrated Trajectory Model (HYSPLIT) trajectory analysis using European Centre for Medium-Range Weather Forecasts Interim Reanalysis (ERA-Interim) 0.75° reanalysis. (**a**) Forward trajectories started at the geographical location of Calbuco at 22 April 2015 18:11 Universal Time (UT), the time of the first eruption, for starting heights above mean sea level (MSL) of 12.8 km to 13.4 km at 100 m intervals. The duration is 187 h and ends at approximately the time of the Buckland Park lidar measurement on 30 April. The source location is marked by a black filled star. Colored filled markers are placed along each trajectory at 00:00 UT each day. An open circle and open square mark the location of the Buckland Park and Kingston lidars, respectively. The center of the open triangle is at the approximate geographical location of the enhanced backscatter marked with an arrow in Figure 2a. (**b**) Backward trajectories started at Buckland Park at 30 April 2015 13:00 UT at heights above MSL of 11.45, 12.15 and 12.75 km (corresponding to layers of enhanced backscatter in Figure 1b). The duration of the run is 187 h and ends at 22 April 2015 18:00 UT. The location of Calbuco is marked by '+'. The lower panels show time series of the height of each trajectory above ground level (AGL).

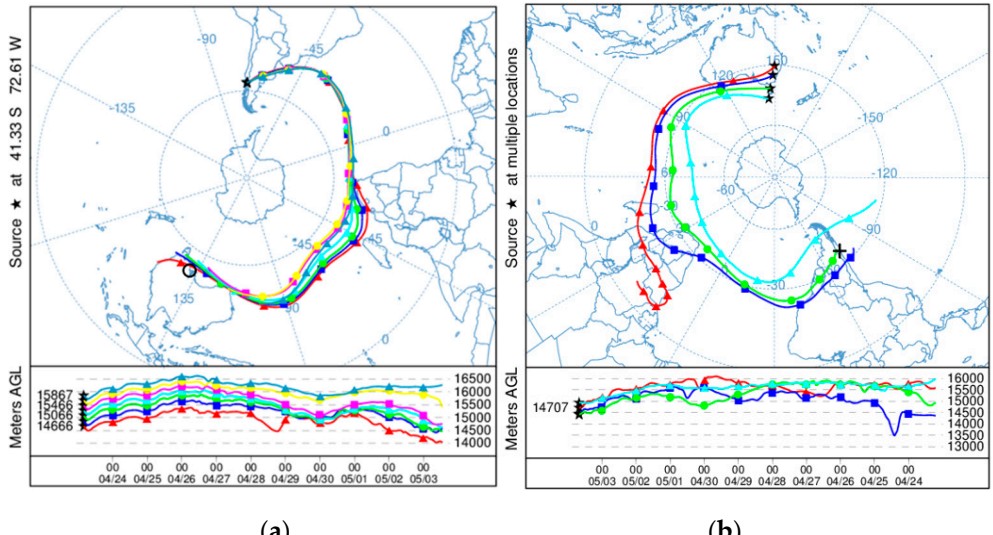

(**a**)                          (**b**)

**Figure 4.** Similar to Figure 3, HYSPLIT trajectory analysis using ERA-Interim 0.75° reanalysis. (**a**) Forward trajectories started at Calbuco (star) at 23 April 2015 04:00 UT (the time of the second eruption) for starting heights above mean sea level (MSL) of 15.3 km to 16.5 km at 200 m intervals. The duration is 249 h and ends at approximately the time of the Buckland Park lidar measurement on 3 May. The location of Buckland Park is marked by an open circle. (**b**) Backward trajectories started at latitude-longitude pairs (30.39° S, 144.77° E), (34.62° S, 148.64° E), (40.66° S, 146.88° E) and (45.94° S, 144.99° E) at heights above MSL of 15.0, 15.0, 14.5 and 15.0 km, respectively (corresponding to geographical locations along the stratospheric aerosol layer in Figure 2b). The duration is 251.5 h and ends at 23 April 04:00 UT. The location of Calbuco is marked by '+'.

Aside from the backscatter layers observed in Buckland Park that are listed in Table 2, we also examined other significant narrow features that are apparent in Figure 1. For example, a local maximum in scattering ratio ~23 km is apparent in both Figure 1b,d, and this had a 1σ lower significance limit that was greater than unity. Above approximately 20 km the stratospheric winds were relatively light and backward trajectories suggest that air masses at these heights generally only moved slowly eastward in the Australian region over the preceding 5–10 days and did not have an obvious connection to the Calbuco eruptions.

### 3.2. Kingston Scattering Ratio

Figure 5 shows scattering ratio profiles obtained with the Kingston lidar on 17 and 22 May. In retrieving these profiles, we initially used $S_{aer}(z) = 50$ sr and then refined the value at specific heights based on optical depth analysis discussed in Section 3.3. On 17 May, a persistent layer was observed with a maximum in scattering ratio at 14.5 km (Figure 5a and Table 2). There was also an enhancement near 17 km which was significant in 500 m vertical resolution analysis. As can be seen in Figure 6a, the other two obvious features at 9.0 km and 11.4 km in Figure 5a varied in brightness during the observation. Weak enhanced scatter was apparent down to at least 11 km, and from 08:40 to 09:15 UT there was an obvious brightening at the base of this layer with individual patches having scattering ratios of up to 4.2. As we discuss in Section 3.4 below, this feature likely contained ice clouds mixed with volcanic aerosol. At 9 km, the bright feature from 09:15 to 09:25 UT had a peak scattering ratio of 8.7, and also as discussed in Section 4.1 had scattering characteristics consistent with an ice cloud.

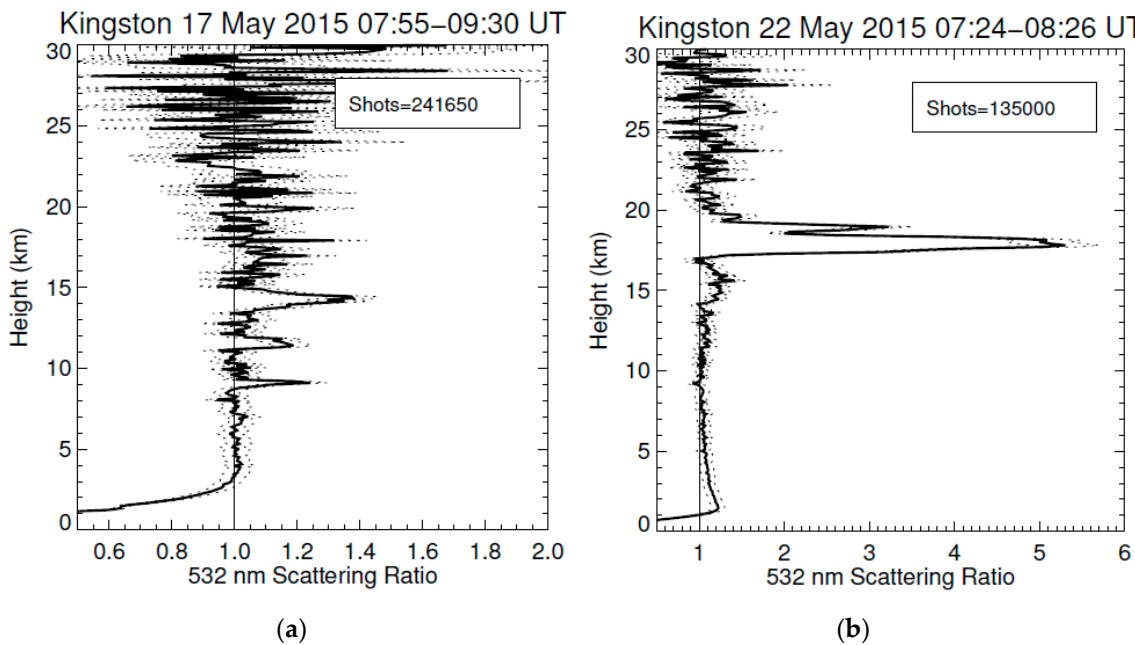

**Figure 5.** Retrieved 532 nm scattering ratio for Kingston lidar measurements on (**a**) 17 May and (**b**) 22 May 2015 at 100 m vertical resolution. Note the different horizontal scales. The solid vertical line marks unit scattering ratio. Thin dashed lines mark 1σ significance limits.

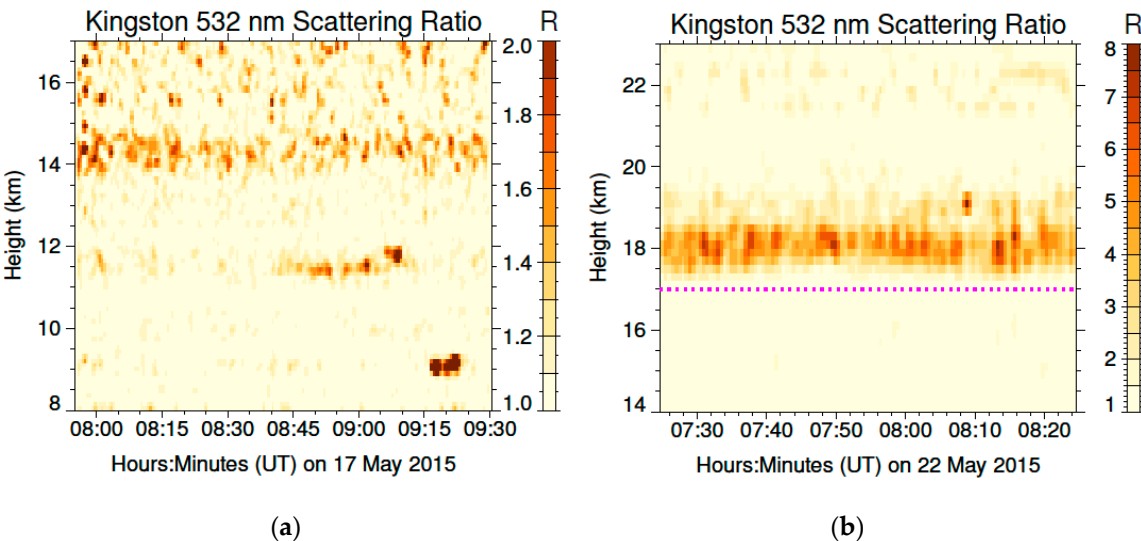

**Figure 6.** Height-time display of 532 nm scattering ratio for Kingston lidar measurements on (**a**) 17 May and (**b**) 22 May 2015. The scattering ratio was retrieved in bins of 100 m × 30 s for (**a**) and 100 m × 30 s for (**b**) and has been smoothed by a Gaussian kernel with a full-width at half-maximum of 2 × 2 bins. In (**b**), the magenta dashed horizontal line is placed at a height of 17 km to aid in examining the vertical variation in the height of the main aerosol layer (centered at 18 km) over time. Note that the vertical and horizontal scales are different between (**a**) and (**b**).

On 22 May, a stronger feature was observed with a peak in scattering ratio at 17.8 km, together with a weaker enhancement at least down to 14 km (Figure 5b). As we discuss in relation to depolarisation measurements in Section 3.4 below, these features represented volcanic aerosols. The strong layer is shown in more detail in Figure 6b. The radiosonde measurements for the 11:15 UT flight on 22 May (~4 h after the lidar measurements, not shown) indicated separate local maxima in wind speed near the heights of peak scattering ratio within the upper and lower parts of the layer, and a shear in wind

direction with increasing height from westerly to west-south-westerly. Figure 6b also suggests that the layer underwent a vertical oscillation of up to ~600 m (3 pixels peak–peak) with a period on the order of 45 min to 1 h during the observation interval. This vertical motion could be due to the passage of high frequency gravity waves. We applied the wavelet gravity wave analysis method used in Murphy et al. [44] to the 11:15 UT flight, and the previous flight at 21:15 UT on 21 May (~7 h before the lidar measurements). For each radiosonde flight, only two wave packets (with downward phase velocity and hence upward group velocity) passed the analysis threshold over the height range 16.5–23 km; both of these packets gave very similar centroid heights when extrapolated to the mid-time of the lidar measurements using the analysed vertical group velocities. These centroid heights were near the upper edge of the main volcanic layer. Table 3 summarises the characteristics of the analysed wave packets. Packets 1 and 2 were of approximately 500–600 m vertical wavelength compared with the ~2 km vertical wavelength of packets 3 and 4; the latter were located approximately 1.5 km higher than the former and had significantly larger temperature and wind perturbations. The estimated horizontal wavelengths of these waves were several thousands of kilometers. Overall, the periods and vertical wavelengths of the wave packets were consistent with the vertical scales and vertical motion apparent in Figure 6b, suggesting that the gravity wave packets may have influenced the vertical displacement of the layer.

**Table 3.** Parameters for analysed gravity wave packets relevant to the Kingston lidar observation on 22 May 2015.

|  | Packet 1 May 21 23:15 UT | Packet 2 22 May 11:15 UT | Packet 3 21 May 23:15 UT | Packet 4 22 May 11:15 UT |
|---|---|---|---|---|
| Extrapolated altitude (km) at mid-lidar observation | 19.4 | 19.2 | 20.4 | 20.6 |
| Ground-based period (h) | 0.7 | 1.1 | 0.6 | 1.4 |
| Vertical wavelength (km) | 0.5 | 0.6 | 1.8 | 2.1 |
| Vertical group speed (m h$^{-1}$) | +38 | +40 | +199 | +63 |
| Peak–peak temperature perturbation (K) | 0.2 | 0.2 | 0.8 | 1.6 |
| Peak–peak zonal wind perturbation (m s$^{-1}$) | 0.6 | 0.1 | 1.9 | 2.0 |
| Peak–peak meridional wind perturbation (m s$^{-1}$) | 0.9 | 0.2 | 3.4 | 2.3 |

The height of the main layer in Figure 6b was consistent with the height of layers of volcanic aerosol subtype identified in contemporaneous CALIOP curtains [36]. These features were also consistent with enhanced 675 nm aerosol extinction in v1.5 analysis (Figure 7) from the Ozone Mapping Profiler Suite (OMPS) instrument on the Suomi National Polar-orbiting Partnership (NPP) satellite [45]. As shown in Figure 7b, the Kingston observations on 17 and 22 May occurred when enhanced OMPS aerosol extinction was apparent up to ~22 km. The Kingston observation on 30 April–1 May and 11 and 28 May did not show any obvious backscatter layers (Table 2), and at these times OMPS showed relatively low extinction in the lower stratosphere suggesting that obvious effects of the Calbuco eruptions were not present. Also of note is that the Buckland Park measurements on 30 April and 3 May were consistent with enhanced OMPS aerosol extinction at the heights of the layers observed by the lidar (Figure 7a). Note that for Figure 7 we have used separation criteria of ±15° in longitude and ±2° in latitude, which equate to ±1220 km in the east-west direction and ±222 km in the north-south direction, respectively. We chose these limits to restrict the sampling to the expected fetch of air around each site over the course of a day. From the radiosonde measurements, the typical wind speed near 15 km was ~20 m s$^{-1}$. An air parcel travelling at this speed would cover ~1730 km over a day, so our longitude range is perhaps overly broad, but necessary to provide a reasonable number of daily samples.

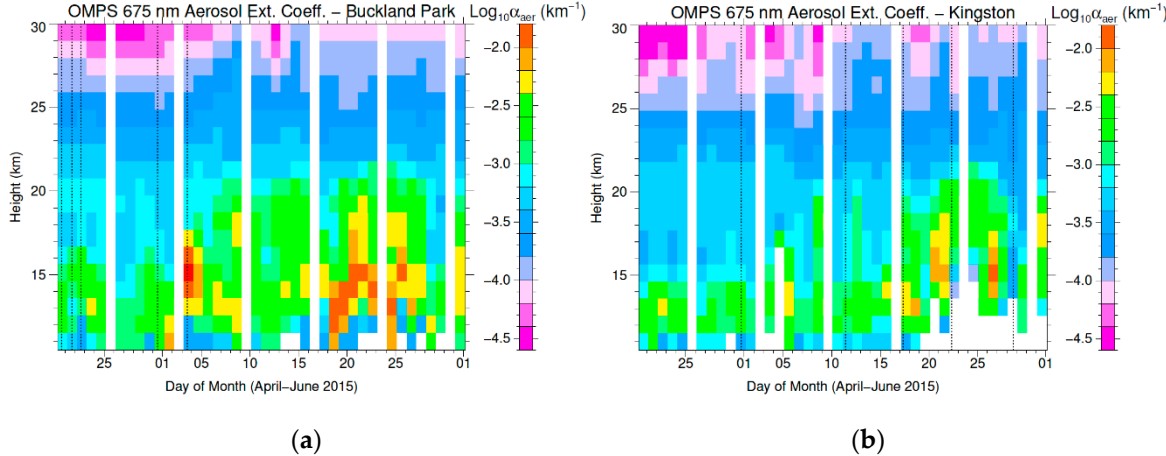

**Figure 7.** Base-10 logarithm of mean daily 675 nm aerosol extinction coefficient profiles from the Ozone Mapping Profiler Suite (OMPS) instrument obtained within 15° in longitude and 2° in latitude of (**a**) Buckland Park and (**b**) Kingston. The vertical dashed lines mark the mid-times of lidar measurements at each site. Regions of no available data are shown in white.

*3.3. Optical Depth and Lidar Ratio*

We used three approaches to obtain the layer-average aerosol transmission $T_c$ of features in time-averaged scattering ratio profiles from the Kingston measurements through methods provided by Uchino et al. [46], Young [47] and Kovalev et al. [48]. The method of Uchino el al. [46] provides

$$T_c(z_l : z_u) = (z_u / z_l) \times \{(P(z_u) \times R(z_l) \times \beta_{mol}(z_l))/(P(z_l) \times R(z_u) \times \beta_{mol}(z_u))\}^{1/2} \tag{11}$$

where $z_l$ and $z_u$ are the heights of the base and top of the layer, respectively. Young [47] gives

$$T_c(z_l : z_u) = \{C_2 \div C_1\}^{1/2} \tag{12}$$

where $C_1$ and $C_2$ are the calibration factors to Equation (1) obtained for small height intervals $(z_b, z_l)$ and $(z_u, z_t)$, respectively, with $z_b$ and $z_t$ being lower and upper height limits, respectively. Both the Uchino et al. [46] and Young [47] methods assume that the regions below and above $z_l$ and $z_u$ have negligible aerosol content, or at least the scattering ratio in these regions is reasonably invariant. We chose the sampling heights where the scattering ratio was uniform as close to the base and top of the main aerosol layers as possible. We modified the Uchino et al. [46] method to obtain mean values of P, R and $\beta_{mol}$ over height ranges $(z_b, z_l)$ and $(z_u, z_t)$, in place of discrete values at $z_l$ and $z_u$. This allowed us to reduce the uncertainty in the derived value of the optical depth in the face of photon noise. These height ranges were chosen as 1 km, except where there was a likely impingement on a nearby layer where the interval was reduced to 0.75 km. Kovalev et al. [48] provide a method to obtain the layer optical depth $\tau_c$ in the situation where a layer is present for only part of the observation time, such as for the case of the short-lived lower layers in Figure 6a. Here

$$\tau_c(z_l : z_u) = 0.5 \times \ln \{(P_1(z_l) \div P_2(z_l)) \div (P_1(z_u) \div P_2(z_u))\} \tag{13}$$

where $P_1$ and $P_2$ are the observed lidar signal at times 1 (without the layer) and 2 (with the layer, respectively. The relationship between the layer optical depth and transmission is

$$\tau_c(z_l : z_u) = -\ln T_c(z_l : z_u) \tag{14}$$

The values of the optical depth allowed us to constrain the lidar inversion by selecting a mean lidar ratio $S_{aer}(z)$ over the height range of the feature such that

$$\int_{z_l}^{z_u} \alpha_{aer}(z)dz = \overline{\tau}(z_l : z_u) \tag{15}$$

where $\overline{\tau}$ is the weighted mean optical depth given by at least two methods.

We applied all three methods for observations on 17 May, and the Uchino et al. [46] and Young [47] methods for 22 May. We did not analyse the Buckland Park measurements using this approach owing to the constraint that we applied to correct for the overlap function. Table 4 lists values of the optical depth obtained from each method (where applicable) and $\overline{\tau}$ obtained for specific layers, along with the selected values of $S_{aer}$. For comparison, values estimated from the OMPS measurements are included. Here, the relevant daily OMPS extinction profiles shown in Figure 7 have been averaged over the height interval of the lidar layers and multiplied by 675/532 = 1.27 to account for the assumed wavelength dependence of the extinction using $\alpha = 1$. There was a reasonably good agreement between the OMPS and Kingston optical depth for the upper layer on 17 May, but a large discrepancy occurred for 22 May. In the latter case, it is evident that such a large difference could only feasibly occur if the satellite was not sampling the strong layer that was measured at Kingston. As noted in the Introduction, Bègue et al. [8] observed the Calbuco aerosol from La Reunion Island. They observed extinction values up to ~0.005 km$^{-1}$ based on an assumed lidar ratio of 60 sr. The average extinction value for the upper layer on 17 May of 0.005 km$^{-1}$ was similar to the upper range observed by Bègue et al. [8]. The value for 22 May of 0.012 km$^{-1}$ was well above the range of their observations, suggesting that the aerosol loading may have been higher at mid-latitudes compared with lower latitudes.

**Table 4.** Evaluations of layer-mean optical depth and lidar ratio at 532 nm from Kingston measurements at 100 m vertical resolution. [1]

| Date (2015) and Time Range (UT) | Height Range (km) | Young Method Optical Depth [2] | Uchino et al. Method Optical Depth [2] | Kovalev et al. Method Optical Depth [2] | Layer-Mean Optical Depth [2] $\overline{\tau}$ | OMPS Optical Depth for 532 nm [2] | Estimated Lidar Ratio $S_{aer}$ (sr) |
|---|---|---|---|---|---|---|---|
| 17 May 07:55–09:30 | 8.6–9.6 | 0.017 (18) | 0.029 (61) | 0.036 (27) | 0.023 (40) | Nil [3] | 36 ± 61 |
| | 10.9–12.3 | 0.017 (12) | 0.018 (23) | 0.011 (14) | 0.014 (17) | 0.0004 (1) | 71 ± 82 |
| | 13.4–15.1 | 0.009 (11) | 0.009 (19) | NA [4] | 0.009 (15) | 0.0070 (3) (13–15 km) | 90 ± 153 |
| 22 May 07:24–08:26 | 14.0–20.5 | 0.078 (27) | 0.080 (39) | NA [4] | 0.079 (34) | 0.0093 (4) (14–20 km) | 86 ± 37 |

[1] Uncertainties are 1σ. [2] Values in parentheses () are the uncertainty in the last digit(s) of the associated measure. [3] No value available as the height range is below the minimum range of the measurements. [4] NA means that the method was not applicable.

The bright scattering feature on 17 May at ~9 km was analysed as having $S_{aer} = 36$, which was consistent with it being a cirrus cloud as discussed below in Section 4.1. The layers above at ~11.6 km and ~14.2 km were analysed as having $S_{aer}$ of 71 sr and 90 sr, respectively, although the uncertainty limits are large because the optical depths of the layers were relatively low. As we discuss in Section 4.2, the uppermost layer had a lidar ratio consistent with backscatter by volcanic aerosol, while as we discuss in Section 4.1, the middle layer was likely a mixture of volcanic aerosol and water-ice. An important aspect to consider for the analysis is the value of the multiple scattering factor η which represents the fraction of photons remaining in propagating beam undergoing multiple scattering in an aerosol layer. This factor is normally only relevant where the scattering layer is optically thick or relatively far from the lidar (e.g., space-based [49]), or the lidar has a relatively large field-of-view. We applied the multiple scattering model of Eloranta [50] to the layers on 17 May and 22 May using the retrieved profiles of $\alpha_{aer}$ as a first-order approximation for the true aerosol extinction and the

field-of-view values for the transmitter and receiver field in Table 1. We estimated that η was at minimum 0.996 for these cases, and therefore we did not need to correct the lidar ratio or depolarisation ratio for multiple scattering.

### 3.4. Depolarisation Analysis

The Kingston observations on 30 April–1 May and 22 May provided measurements of depolarisation. For comparison with other published measurements, we calculated the particle linear depolarisation ratio $\delta_p$ using Equation (20) of Freudenthaler et al. [51] as

$$\delta_p(z) = ((1 + \delta_m)\, \delta_v(z) \times R(z) - (1 + \delta_v(z))\, \delta_m)\, / ((1 + \delta_m)\, R(z) - (1 + \delta_v(z))) \tag{16}$$

where $\delta_m$ is the depolarisation ratio of atmospheric molecules, which is taken as 0.37% (as appropriate for our choice of interference filter listed in Table 1 using Behrendt and Nakamura [52]). In Table 5, we provide layer mean and maximum values of $\delta_p$ spanning the altitude range of the two main aerosol layers observed on 22 May; standard errors are provided for mean values.

**Table 5.** Evaluations of particle depolarisation ratio for the Kingston measurements on 22 May using 100 m vertical resolution analysis.

| Height Range (km) | Mean Particle Linear Depolarisation Ratio $\delta_p$ (%) | Maximum Particle Linear Depolarisation Ratio $\delta_p$ (%) | Estimated Ash Backscatter Fraction |
|---|---|---|---|
| 15.0–16.0 | 27 ± 9 | | 0.80 ± 0.22 |
| 17.2–18.5 | 13.9 ± 1.9 | 25 ± 29 (17.7 km) | 0.44 ± 0.27 |
| 18.5–19.1 | 26.0 ± 7.0 | 54 ± 72 (18.6 km) | 0.77 ± 0.18 |
| 17.2–19.1 | 18.0 ± 3.0 | | 0.56 ± 0.20 |

The depolarisation measurements for the 30 April–1 May observation did not show any noticeable enhancements, except for a deepening cloud layer below 12 km from 04:00 UT on 1 May. These clouds had mean volume depolarisation ratios of ~0.3 to 0.5 typical of cirrus [53,54]. The mean profile of the volume depolarisation ratio for 22 May is presented in Figure 8. The formal uncertainty in the individual stratospheric measurements is relatively large for this short observation (up to 21% at the 1σ level), but the presence of increased depolarisation can be seen at the heights of the main enhanced backscatter features in Figure 6b. Based on our depolarisation calibration, we estimate the value of possible systematic bias due to misalignment of the analyser polarization planes was 5% (1σ); this is less than the baseline level of the depolarisation apparent in the free troposphere in Figure 8 and similar in magnitude to calibration biases analysed by Belegante et al. [31]. Note that this bias was not incorporated in the formal uncertainty derived from photon counting statistics shown in Figure 8, and that the level of the bias is generally small in comparison with the formal uncertainty at stratospheric heights.

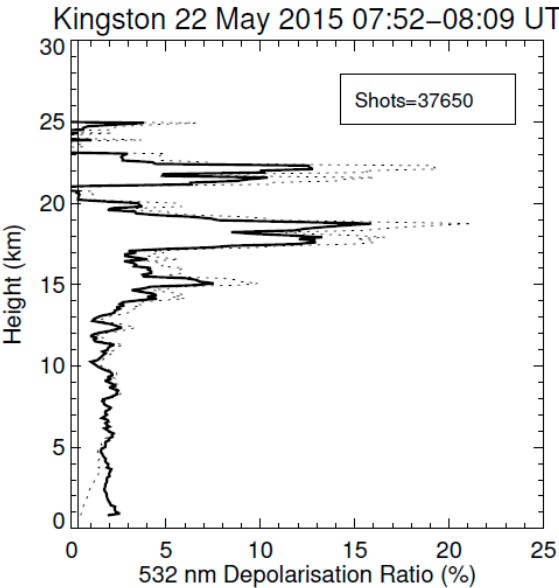

**Figure 8.** Mean profile of 532 nm linear volume depolarisation ratio $\delta_v(z)$ for Kingston measurements on 22 May 2015 for heights 1–25 km. The depolarisation ratio was retrieved at 100 m vertical resolution and has been smoothed with a 5-point running mean. The solid vertical line marks the estimated linear depolarisation for molecular backscatter (0.37%) as discussed in Section 3.4. The thin dashed line shows the profile of the 1σ uncertainty based on photon counting statistics and is shown on the same scale as the horizontal axis.

The maximum in $\delta_v$ was 16% at 18.6 km, and this occurred near the lower edge of the upper semi-detached part of the main layer. Higher values of the mean particle depolarisation ratios were observed in the upper part of the layer (Table 5). Light penetrating into a scattering layer will be depolarised by multiple scattering, with the observed depolarisation increased by a factor of 1/η. However, given that η discussed in Section 3.3 was close to unity, multiple scattering cannot account for the observed depolarisation difference between the upper and lower parts of the layer.

Using Equation (4) and following Tesche et al. [55] and Ansmann et al. [56], we provide in Table 5 estimates the fraction of backscatter from ash in separate parts of the layer, where it is assumed that ash and non-ash (fine mode particles, of likely sulfate composition) have particle depolarisation ratios of 0.36 and 0.01, respectively. As can be seen from Table 5, there is the indication that the upper part of the layer was more dominated by ash backscatter than for the lower part and that overall the layer was ash-rich (56% ash backscatter fraction). We may have expected an ash-dominated layer to be located below the sulfate layer on the basis of expected particle size. The differences in the ash fraction over the layer could simply represent separate filaments of material that originated from the separate eruptions. However, it is also possible that the action of mesoscale waves and gravity waves acting on particle aggregation and diffusion subsequently stratified and processed the layers into different compositions [2,57,58]. Note that Figure 6b shows variability in the brightness of the lower part of the main layer. We suspect that this is related to changes in the number density of the fine-mode particles in the layer, but were unable to test this by examining the dependence of the backscatter coefficient with the volume depolarisation as is commonly done to discriminate aerosol properties (e.g., [59]). This was because of the very low signal-to-noise on the individual depolarisation measurements at the timescale of the variability in Figure 6b.

The relatively weak scattering layer centered at 15.5 km apparent in Figure 6b, which we presume was also of volcanic origin by virtue of its relatively high depolarisation compared with the aerosol at lower altitudes (Figure 8), gave a high ash backscatter fraction that was similar to the value for the upper part of the main layer. Note that Figure 8 shows enhanced depolarisation from 21 to 23 km,

where accompanying weak backscatter can be seen in Figure 6b. We could not determine the particle depolarisation with any accuracy at these heights owing to the level of photon noise.

CALIPSO overpasses near Tasmania were available at approximately 05:00 UT (day) and 16:00 UT (night), both of which were west of Kingston at distances of 550 km and 750 km, respectively. HYSPLIT analysis suggests that the air-mass at 19 km height arrived at Kingston from the west-southwest direction, and was generally in a similar location to the CALIOP measurements between latitudes of 43.5° S and 44.8° S. Daytime CALIOP depolarisation retrievals are problematic for weak scattering layers and it was not possible to obtain a meaningful estimate for the 05:00 UT pass. Outlier depolarisation values, both negative and positive, are also present in the nighttime measurements, and applying a limit on the quoted formal uncertainty in depolarisation of 50% produced a robust frequency distribution of values. Applying this restriction to the 16:00 UT pass, the mean 532 nm particle linear depolarisation obtained directly from the v4.20 CALIOP aerosol product over the selected latitude range was 14.1 ± 0.9%. The CALIOP particle linear depolarisation ratio is a post-extinction quantity calculated from polarization components of the particle depolarisation coefficient but is not adjusted for multiple scattering [60], and is equivalent to the quantity provided in Equation (4). The corresponding CALIOP mean 532 nm scattering ratio was 4.3 ± 0.2. This analysed layer had a height range of 17.0–20.4 km, which was similar to the main layer observed at Kingston (Figure 5b and Table 2). For Kingston, the mean values of particle linear depolarisation ratio and scattering ratio over the height range of the layer were 18.0 ± 3.0% (Table 5) and 3.4 ± 0.3, respectively, and broadly consistent with CALIOP. For the 05:00 UT CALIOP pass, the mean scattering ratio was 7.1 ± 0.3 for the analysed layer in the height range 18.3–19.0 km, which generally covered the narrow upper layer observed at Kingston. Similar mean depolarisation and scattering values were obtained when the average for the CALIOP nighttime pass was extended to latitudes 50° S to 35° S, and in this case with more measurements available, the frequency distribution (not shown) had a clear peak at ~10% and a weaker peak at ~25% suggesting two broad populations of particles, with a higher mean particle depolarisation at the upper heights which was consistent with the Kingston observations (Table 5). The aerosol layer near Tasmania appeared inhomogeneous in the CALIOP curtains (e.g., [61]), and so the differences between the layer average values presented above could simply be due to variations within the layer, and the time and distance separations between the measurements. Forward trajectories from the location of the analysed CALIOP aerosol layer for the 05:00 UT measurements passed approximately 100 km south of Kingston approximately 14 h after the lidar observations. For the 16:00 UT pass, a similar distance separation occurred approximately 22 h after the Kingston observations. Despite these time differences, the coincidences discussed above were the closest we could find for the 22 May Kingston measurements.

## 4. Discussion

### 4.1. Synergy of Lidar and Radiosonde Measurements

In Figure 9, we compare the Buckland Park lidar profiles with temperature and water vapor measurements from contemporaneous Vaisala RS92 radiosonde profiles obtained from Adelaide Airport. We were interested to determine the possibility of involvement of ice clouds in the observed layers given their proximity to typical tropopause heights, and the detection of the ash-cloud mixture above Kingston on 17 May.

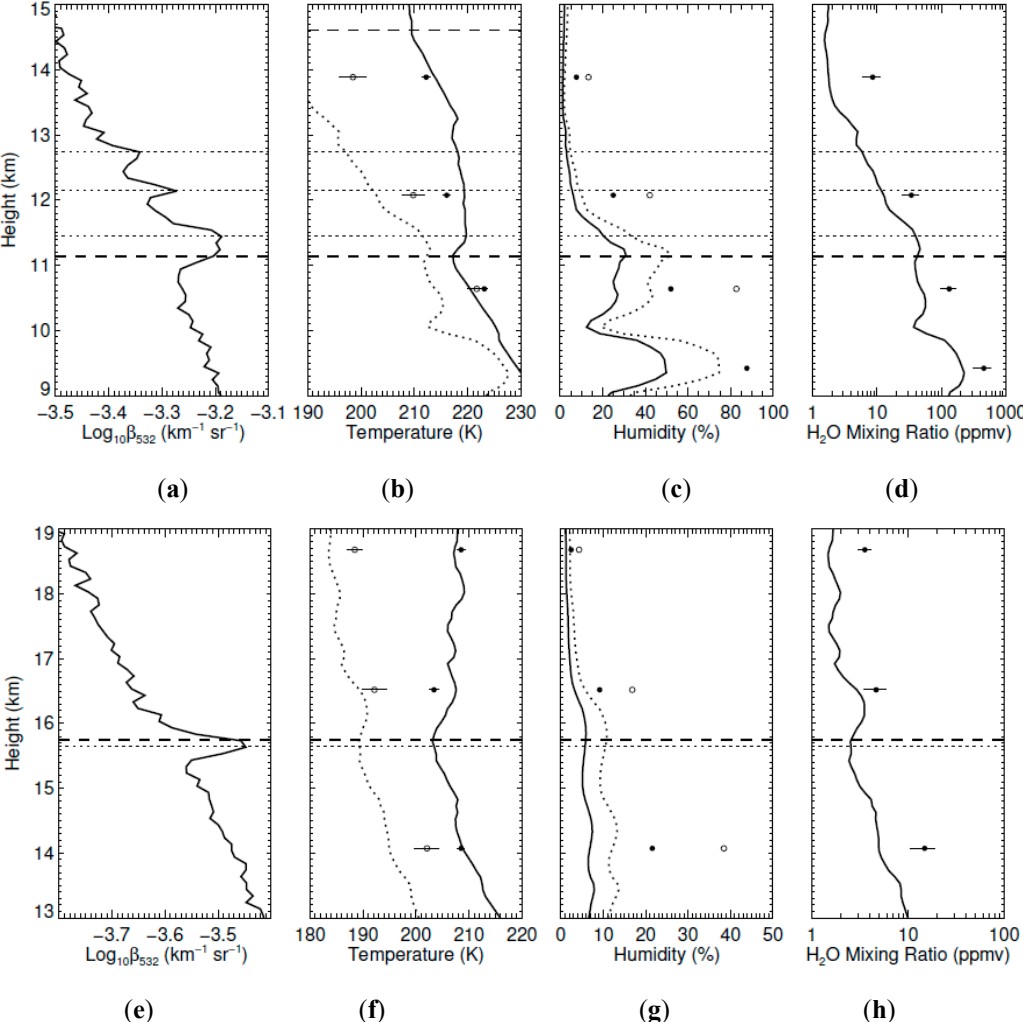

**Figure 9.** Comparisons of Buckland Park lidar, radiosonde and AIRS measurements for 30 April
(panels (**a**) to (**d**)) and May 03 (panels (**e**) to (**h**)) 2015. Panels (**a**) and (**e**) show the mean total attenuated
lidar backscatter. The thin dotted horizontal line marks the height of peak backscatter for identified
layers. The thick dashed horizontal line marks the height of the first thermal tropopause obtained
from the associated radiosonde measurements. Panels (**b**) and (**f**) show temperature (solid profile) and
water-ice frost point temperature (dashed profile) from radiosonde measurements. Filled and open
circles with $1\sigma$ uncertainty limits show AIRS measurements of temperature and water-ice frost point
temperature, respectively. Panels (**c**) and (**g**) show radiosonde profiles and AIRS measurements of
humidity (solid profile and filled circles, respectively) and humidity over ice (dashed profile and open
circles, respectively). Panels (**d**) and (**h**) show the water vapor mixing ratio from radiosonde and AIRS
measurements (solid profile and filled circles with $1\sigma$ uncertainty limits, respectively). The times of the
radiosonde and AIRS measurements are noted in Table 2. In panel (**b**), the thin dashed horizontal line
marks the second thermal tropopause. The vertical resolution is 200 m.

On 30 April, Figure 9a indicates that the lowermost layer in the lidar measurements was located
at or slightly above the first thermal tropopause (the height of which is based on [62]). This layer
and the other two above it spanned a range of height where the temperature was relatively constant
at approximately 220 K (solid black line in Figure 9b). The measured relative humidity over water
and ice (solid and dashed black lines, respectively in Figure 9c) show a local peak at the tropopause.
Radiosonde humidity measurements are problematic at low temperatures and dry atmospheres. The
profiles shown here are corrected for dry bias according to Miloshevich et al. [63]. For comparison,
we show humidity over water and ice derived from AIRS water vapor mixing ratio measurements

coincident with the lidar observation (filled and open circles, respectively, in Figure 9c). The agreement between the radiosonde and AIRS measurements varies with height with differences up to a factor of ~2. However, the variation of humidity with height is generally consistent between the two types of measurements. The absolute water vapor mixing ratio obtained from the radiosonde and AIRS measurements (solid line and filled circles, respectively, in Figure 9d) generally decreased linearly with pressure across the aerosol layers. Around the tropopause, the humidity over ice as determined by the radiosonde measurements was well below saturation, indicating that ice cloud formation was unlikely [64]. This is also suggested in Figure 9b, where the water-ice frost point temperature (dashed line) is colder than the ambient temperature. However, the situation based on the AIRS measurements is less clear, with the humidity over ice approaching saturation for the measurement at ~10.7 km. For the lidar observation on 3 May (Figure 9e,f), the backscatter layer at 15.7 km, while straddling the thermal tropopause, was evidently dry in the coincident radiosonde and AIRS measurements, and as a result we suggest that the possibility of ice cloud involvement in the layer appears unlikely.

The potential temperature ($\theta$) at the height of each inferred volcanic layer is provided in Table 2. Under the assumption that the aerosols were moving on isentropic (i.e., constant $\theta$) surfaces, there was possibly a connection between some layers. For example, the Buckland Park layers on 30 April and 3 May, and the Kingston layers at 11.4 km and 14.5 km on 17 May, all being near the tropopause, had similar $\theta$. The weak layer at 17.2 km on 17 May had similar $\theta$ to the main layer on 22 May. Note that the layer on 3 May and the main layer on 17 May were close to the level of the advected potential vorticity maps shown in Figures 11 and 12 of Bègue et al. [8], and thus appear to have had connections to measurements made at La Reunion.

We also note that we applied the gravity wave analysis outlined in Section 3.2 to the nearest radiosonde flights to the 30 April and 3 May lidar observations but did not identify any packets of short vertical wavelength that appeared to be associated with the aerosol layers.

In Figure 10 we show scenes around Buckland Park coincident with the lidar measurements of the brightness temperature differences between two infrared channels of the MODIS instrument on the Terra satellite. The brightness temperature difference for the selected channels highlights ice clouds, which appear bright. For the scene of 30 April (Figure 10a), the closest obvious cloud feature is near to the location of the AIRS measurement (blue open triangle) made at 16:29 UT, 23 km from the lidar.

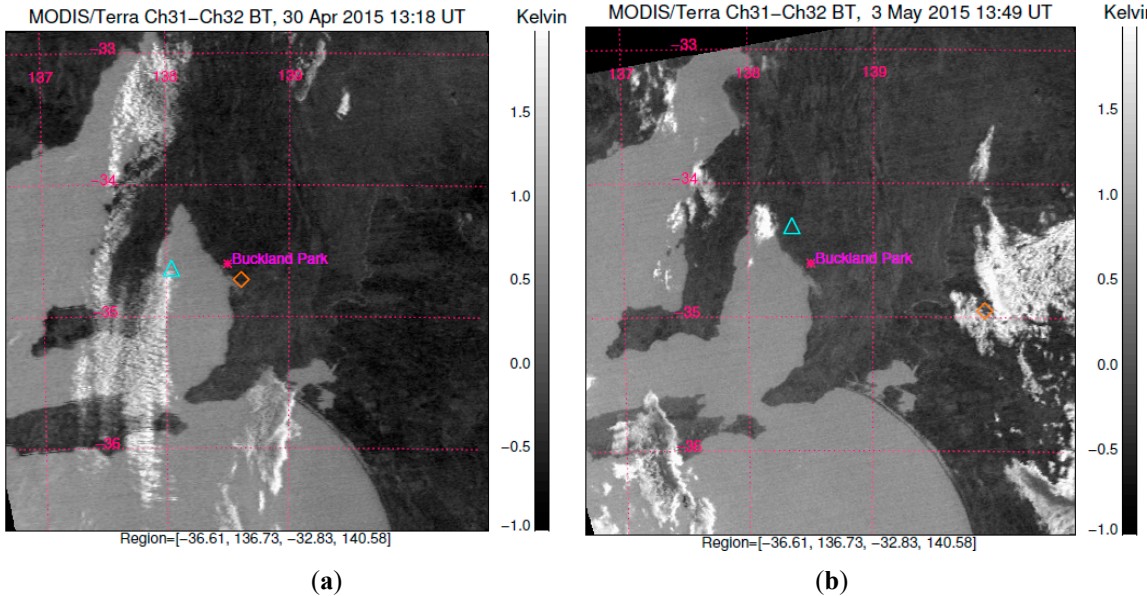

**Figure 10.** Brightness temperature difference between Moderate Resolution Imaging Spectroradiometer (MODIS) infrared channels 31 (11 μm) and 32 (12 μm) for the nearest overpasses to the Buckland Park lidar measurements on (**a**) 30 April and (**b**) 3 May 2015. The orange open diamond marks the location of the 11:00 UT radiosonde launch for each day when it reached a height of approximately 12 km for (**a**) and 15 km for (**b**) (corresponding to the approximate heights of the identified volcanic aerosol layers in Figure 1). The light blue open triangle indicates the location of the nearest AIRS granule to the lidar site for overpasses on 30 April at 16:29 UT for (**a**) and 3 May at 15:27 UT for (**b**). The scenes span approximately 355 km (E-W) by 420 km (N-S).

The radiosonde measurement at the tropopause (orange open diamond in Figure 10a) was situated close to Buckland Park where the scene appeared free of cloud. While we cannot rule out the possibility of involvement of ice aerosols in the lowermost backscatter layer measured by the lidar based on the radiosonde profile, this seems unlikely based on the absence of obvious cloud in the vicinity of Buckland Park and the relatively low value of the humidity over ice in the radiosonde profile. The clouds could still be optically too thin to detect in this MODIS images. The MODIS image for 3 May (Figure 10b) indicates that the radiosonde measurement at the height of the main lidar backscatter layer was potentially close to a cloud deck, but based on examination of the radiosonde humidity profile, this deck was evidently at a height range of 3–5 km.

We also examined radiosonde and AIRS measurements contemporaneous with the Kingston measurements (Figure 11). The volcanic layer at 14.6 km on 17 May was situated approximately 1500 m above the tropopause (Figure 11a), where conditions were evidently dry (Figure 11b–d) as similarly observed for the layers observed at Buckland Park. On 22 May the three layers identified in Figure 11e (horizontal dashed lines) also showed minimal water vapor content (Figure 11f–h).

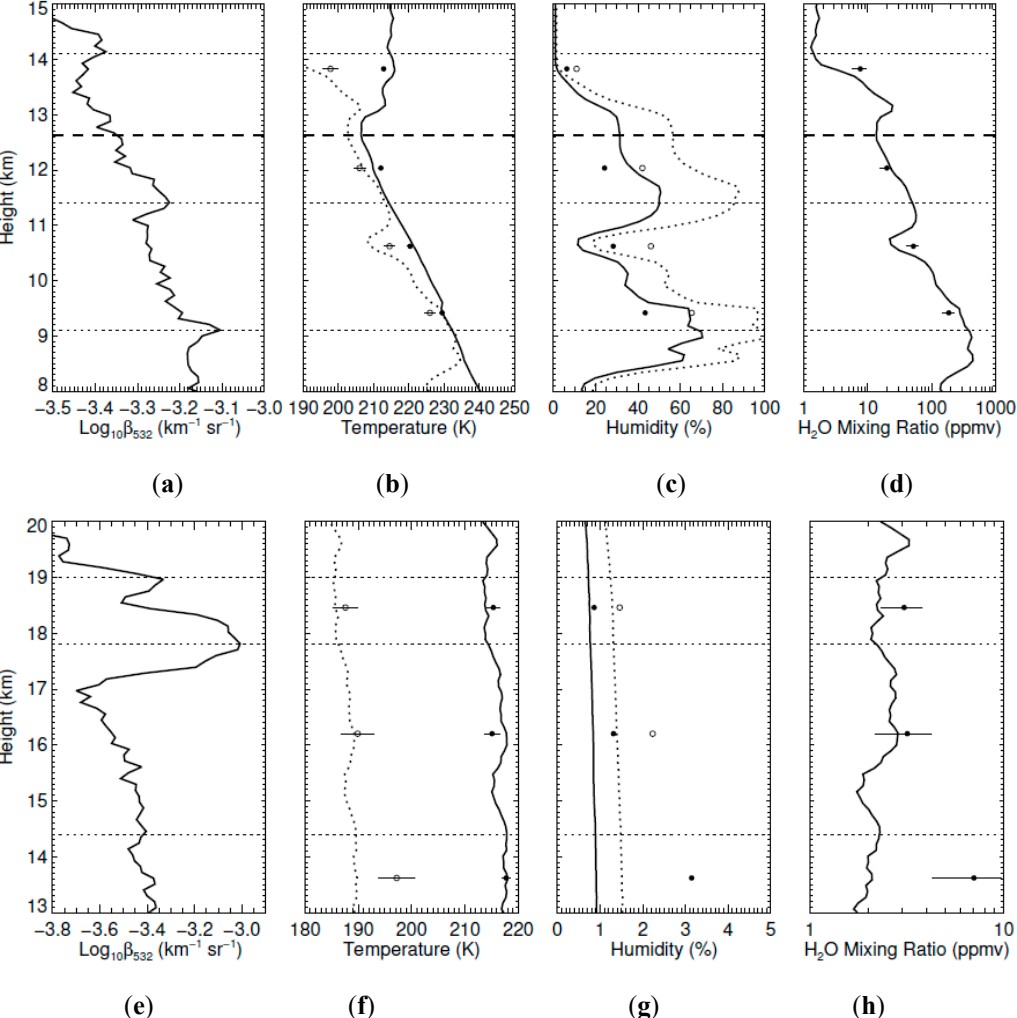

**Figure 11.** Similar to Figure 9, but for comparisons of Kingston lidar, radiosonde and AIRS measurements for 17 May (panels (**a**) to (**d**)) and 22 May (panels (**e**) to (**h**)) 2015. The radiosondes were launched from Hobart Airport, approximately 25 km from the lidar site. The separation of the radiosonde measurements from the lidar measurements in distance (bearing) and time was as follows; 17 May at 12 km height: 45 km (ENE), +3.2 h (for panels (**b**) to (**d**)); 22 May at 18 km height: 92 km (NE), +4.3 h (for panels (**f**) to (**h**)). The associated AIRS measurements were obtained on 17 May at 04:34 UT, 26 km NNW of Kingston (for panels (**b**) to (**d**)), and 22 May at 04:52 UT, 29 km WSW of Kingston (for panels (**f**) to (**h**)). The vertical resolution is 200 m.

As can be seen in Figure 11a, the two ephemeral features on 17 May (discussed in Section 3.2 in relation to Figure 6a) were situated below the tropopause; at the heights of these features the radiosonde-derived humidity over ice was close to saturation, indicating that these features were potentially ice clouds. As we noted in Section 3.4, the lowermost layer at 9 km had an inferred lidar ratio of $S_{aer} = 36$ sr which is consistent with the expected range for cirrus clouds of 20–40 sr found by Chen et al. [45].

For the bright region of the middle layer at 14.6 km, $S_{aer} = 71$ sr which is consistent with volcanic aerosol [28], but less than the lidar ratio for the upper layer ($S_{aer} = 92$ sr), and higher than for cirrus [45]. Considering the relatively high humidity over ice at the height of the layer, and the appearance of weak aerosol scattering in its vicinity, we suggest that the feature represented a mix of volcanic aerosol and ice, with water vapor potentially undergoing ice nucleation because of the presence of the aerosol. Close inspection of the scattering ratio time-height data for this layer shows evidence of thin layers in

the brightest cloud features descending at speeds of up to ~5 m s$^{-1}$, reminiscent of cirrus fallstreaks. This descent can be discerned in Figure 6a for the bottom edge of the bright cloud visible over the period 09:05–09:10 UT.

Interestingly, our gravity wave analysis for the 11:15 UT radiosonde profile showed wave packets with a downward phase (upward group) velocity that would have been located at approximately 9.7 km and 12.9 km at the time of the lidar observation; this would place the wave packets at the upper edge of the cloud features. These waves had intrinsic periods of ~50 min. The temperature and wind perturbations of the packets were; lower packet: 0.4 K and 0.7 m s$^{-1}$, respectively; upper packet: 1.3 K and 2.0 m s$^{-1}$. While these perturbations were relatively modest, they may have aided the formation of heterogeneous ice nuclei and hence the growth of cloud particles [65], particularly if the wind perturbation assisted the transport of volcanic aerosols into the region of high water vapor saturation.

Volcanic aerosol-induced cirrus clouds have been observed in association with a variety of eruptions. Sassen [66] suggested that supercooled liquid droplets observed in cirrus results from the presence of aqueous ammonium sulfate associated with volcanic aerosols. Campbell et al. [67] observed cirrus clouds embedded in a layer of volcanic aerosol from the 2008 Kasatochi eruption in Alaska. They hypothesized that cloud formation was driven either by homogeneous freezing of aqueous sulfate solutions, or by heterogeneous droplet activation involving the volcanic particles. Shibata et al. [68] observed liquid volcanic aerosols in association with tropical cirrus layers. Friberg et al. [69] observed a decrease in mid-latitude cirrus cloud reflectance observed by satellite sensors following an increase in volcanic aerosol from several tropical eruptions. They suggested that the dimming was due to deactivation of the nuclei responsible for homogeneous freezing by the presence of volcanic sulfate aerosol.

Rolf et al. [70] observed ice formation induced by ash from the Eyjafjallajökull eruption and found that to reproduce the microphysical properties of the observed cloud layer required efficient and enhanced seeding by ice nuclei. Using results from Seifert et al. [71] and Ansmann et al. [72]. Rolf et al. [70] estimated the number of potential ice nuclei present in the ash plume using the observed aerosol extinction. For the height range 10.9–12.3 km which covered the region containing the bright backscatter we obtained a mean aerosol extinction coefficient of 0.0038 ± 0.0005 km$^{-1}$ before the appearance of the cloud, 0.0102 ± 0.0020 km$^{-1}$ (layer-average optical depth ~0.014) during the cloud, and 0.0072 ± 0.0010 km$^{-1}$ after the cloud (assuming $S_{aer}$ = 71 sr). Using the conversion factor to particle concentration provided by Seifert et al. [71] we estimate the mean particle concentration in the region before the cloud of 2.9 ± 1.9 cm$^{-3}$. If we assume that 1% of the aerosols were the most efficient in forming ice nuclei (IN) through heterogeneous nucleation as found by Steinke et al. [73] in a laboratory study of Eyjafjallajökull ash, the IN-forming particle concentration before the appearance of the cloud was estimated as 0.029 ± 0.0019 cm$^{-3}$. This value is similar to and larger than cirrus-active concentrations of heterogeneous nuclei measured in the western United States by DeMott et al. [74] for humidities > 75% and similar temperatures (< ~0.008 cm$^{-3}$ as shown in their Figure 2). Overall it appears feasible that the concentration of IN provided by the background aerosol, which we presume was volcanic based on the lidar ratio of the layer above which appears to have an association with the cloud-forming region, was sufficient to cause the formation of ice crystals in the layer. Unlike Rolf et al. [70] we did not see a noticeable decrease in the inferred particle concentration following the appearance of the cloud which would signify the removal of particles, but we only had a very short observation window (15 min) during this period and it may have been that cloud particles persisted for some time. We note also that based on the Kingston nighttime observations on 30 April, the pre-eruption mean background extinction over the height range was evidently negligible (mean −0.0002 ± 0.0003 km$^{-1}$ assuming a $S_{aer}$ = 50 sr).

The mean extinction over the height range of the lowermost cirrus cloud was 0.007 ± 0.001 km$^{-1}$. Using relations in Heymsfield et al. [75] together with the observed temperature at the cloud layer (Figure 11a) we obtained an effective mean particle diameter $D_e$ of 165 μm and a mean ice water content (IWC) of 0.36 ± 0.05 g m$^{-3}$. For the cloud embedded in the volcanic aerosol, assuming that the

extinction is dominantly from the cloud particles, we obtained $D_e$ = 82 μm and IWC = 0.25 ± 0.05 g m$^{-3}$. The fallstreaks speeds for this cloud noted above are consistent with a sedimentation speed for spherical particles with a diameter of up to ~200 μm, assuming no vertical wind. As determined from Schumann et al. [76] hexagonal plates with a volume-equivalent diameter D of 200 μm have $D_e$ = 0.42 D = 84 μm; this is fully consistent with the expected value of $D_e$ obtained above, which is based on temperature alone. In the event that hexagonal crystals formed, it possible that specular reflection could have enhanced the observed backscatter as the lidar was directed to the zenith and such crystals would tend to fall with their long axes horizontally aligned.

*4.2. Comparison with Other Volcanic Aerosol Measurements*

Prata et al. [28] provided a comprehensive evaluation of lidar ratio and depolarisation for the volcanic plumes associated with the Puyehue, Kasatochi and Sarychev eruptions using CALIOP data. Their analysis assumed η = 0.90 ± 0.05. They observed a wide range of mean volume particle linear depolarisation ratios (equivalent to the layer-mean of $\delta_P$ given by Equation (4) multiplied by η), ranging from 5 ± 4% for Sarychev, 9 ± 3% for Kasatochi, to 33 ± 3% for Puyehue. Nakamae et al. [14] reported values in the range 20–30% for Puyehue. For comparison, our value was 18 ± 3%. Judging from the distributions of $\delta_P$ provided by Prata et al. [28] in panels a–c of their Figure 2, our observations were at the upper end of the distribution for Kasatochi and less than the lower end of the distribution for Puyehue. Ansmann et al. [77] and Gasteiger et al. [78] observed much higher $\delta_P$ values of 35–40% for the Eyjafjallajökull eruption.

The lidar ratio values measured by Prata et al. [28] were relatively consistent across the three volcanic events that they studied; we obtained a weighted mean of 66 ± 9 sr from their data. This is compared with our weighted mean value of 86 ± 111 obtained for the combination of the upper layer on 17 May and the full layer on 22 May. Our mean is approximately at the upper quartile of the values observed by Prata [28] judging from panels a-c in their Figure 2, but our uncertainty limits are broad. Ansmann et al [77] found values of 55 ± 5 sr (Munich) and 60 ± 5 sr (Leipzig) for Eyjafjallajökull aerosols; the 1σ lower bound of our most precise measurement on 22 May (Table 4) is consistent with these values, but its mean (86 sr) is also obviously high in comparison. We note than the time evolution of the lidar ratio provided by Prata et al. [28] in their Figure 8a–c suggests an increase as the layers age. This could be due to the reduction of the effective ash particle size or sphericity because of, for example, sedimentation. Our measurements were made 25–30 days after the Calbuco eruptions; this interval is not covered by the analysis of Prata et al. [28] which spans up to 16–24 days after the events they studied. We also note that Kokkalis et al. [27] demonstrated changes in depolarisation ratio, lidar ratio and Ångström exponent within Eyjafjallajökull volcanic layers during their transit to the Eastern Mediterranean. They observed 532 nm lidar ratios ranging from 44 ± 8 sr to 88 ± 7 sr in the troposphere, which when taken with other measurements that they compared with, generally indicated that higher values were observed as the plume aged. Quantifying the ageing of volcanic plumes is relevant to assessing their climate forcing and understanding the evolution of the aerosol lidar ratio and depolarisation is useful in this regard. We encourage further assessment of the properties of the Calbuco aerosols to help put our measurements into a more detailed context.

## 5. Conclusions

We observed aerosol layers from the April 2015 eruption of the Calbuco volcano in Chile with a Raman/Mie/Rayleigh lidar at Buckland Park, South Australia, and a depolarisation/Mie lidar at Kingston, Tasmania, during April and May 2015. Measurements at Buckland Park on 30 April and 3 May detected discrete aerosol layers at and near the tropopause. Based on backward air parcel trajectory analysis and CALIOP space-based lidar data, these layers likely originated from the major eruptions on 22–23 April. Aerosol layers from the eruptions were subsequently detected during two of five measurement sessions with the Kingston lidar.

Measurements at Kingston on 22 May provided an estimate of the mean particle linear depolarisation ratio of 18.0 ± 3.0% within the main observed layer, which was consistent with the range of CALIOP depolarisation values obtained in a wider region of the aerosol layer near Tasmania. This relatively high depolarisation value suggests that particles in the layer were dominated by backscatter from ash rather than sulfate particles, although there was evidence of a greater backscatter by sulfate particles in the lower part of the layer. The observed mean particle depolarisation was generally higher than for the Kasatochi and Sarychev eruptions [28], less than for the Puyehue eruption [14,28] and much smaller than for the Eyjafjallajökull eruption in 2010 [68,69].

Layer-average values of the optical depth for discrete volcanic layers were used to infer the lidar ratio for Kingston observations on 17 and 22 May. There was reasonable agreement between the lidar-inferred optical depth and the value obtained from measurements by the OMPS instrument on the Suomi NPP satellite on 17 May. The large difference for the comparison on 22 May most obviously was a consequence of the spatial inhomogeneity of the volcanic aerosol clouds. Estimates of the lidar ratio generally had large uncertainties because of the relatively low optical depth of the layers. The most precise measurement was for the strong layer on 22 May which had a lidar ratio of 86 ± 37 sr. A similar mean value was observed on 17 May (92 sr). These values were generally at the upper range observed for other volcanic events in the past decade [8], which could point to a difference in the characteristics of the particular observed plumes, or the Calbuco aerosol in general, in comparison to other volcanic events, or may at least be partly be due to ageing of the plume.

The lidar ratio estimates showed that bright backscatter embedded within an extended region of diffuse aerosol below a volcanic layer on 17 May likely represented clouds of ice crystals. The formation of these clouds, at least for those embedded in the volcanic layer, was potentially aided by the presence of the volcanic aerosol serving as ice nucleation centers. Fallstreaks in the layers suggest the presence of large particles consistent with ice crystal in the form of hexagonal plates.

We provided evidence that gravity wave packets were associated with some layers, particularly for the strong layer on 22 May, and that on three occasions layers were closely associated with the tropopause. This highlights that dynamical factors potentially played an important role in determining the vertical structure of these layers.

Overall, these measurements provide some new insights into the properties of the Calbuco aerosol at mid-southern latitudes. Follow-up analysis of the general properties of the evolving aerosol, particularly along the lines provided by Prata et al. [28] using the combination of CALIOP lidar and satellite infrared sensing of volcanic gases, would help put the measurements provided here into a broader context.

**Author Contributions:** A.R.K. coordinated the preparation of the analyses, led the development of the instrumentation and measurements at Kingston and prepared the main part of the text. D.J.O. contributed to the text and led the instrument development and measurement campaign at Buckland Park (BP). A.D.M. contributed to review of the text and assisted with the BP measurements. I.M.R. contributed to the text and was responsible for the funding and development of the BP lidar and the measurement campaign. L.V.T. helped to develop the BP lidar instrumentation and assisted with measurements. S.P.A. contributed to the text and assisted with the Kingston campaign. All authors have read and agreed to the published version of the manuscript.

**Funding:** This research is supported by the Australian Research Council (ARC) under grants DP0450787, LE0560872 and DP0878144 and by the University of Adelaide ARC Small Grants Scheme. L.V.T. was supported by an Australian Postgraduate Research Award. The involvement of I.M.R. was supported by ATRAD Pty Ltd. This work was also supported by Australian Antarctic Science projects AAS-737 and AAS-4292.

**Acknowledgments:** We thank Blair Middlemiss of the University of Adelaide for invaluable assistance with the construction, setup and operation of the lidar instrumentation at Buckland Park. We also thank Steven Whiteside, Eric King (deceased), Lloyd Symons, Peter de Vries, Chris Richards and Chris Young of the Australian Antarctic Division for their considerable assistance with the development, construction and operation of the Kingston lidar. CALIPSO browse images were obtained from https://www-calipso.larc.nasa.gov/products/lidar/browse_images/production/. AIRS, OMPS and MODIS data were downloaded from the NASA Goddard Earth Sciences Data Information and Services Center (GESDIC; https://disc.gsfc.nasa.gov). Radiosonde data were provided by the Australian Bureau of Meteorology. We thank two anonymous reviewers for helping to improve the content of this manuscript.

**Conflicts of Interest:** The authors declare no conflict of interest.

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
