# Peer review of "Australian Lidar Measurements of Aerosol Layers Associated with the 2015 Calbuco Eruption"

_atmosphere, doi:10.3390/atmos11020124_

Round 1
Reviewer 1 Report
The authors are presenting a synergy of measurements (among them also lidar profiles), observing volcanic layers from Calbuc eruption in 2015, over Australia. The manuscript is well written and holds strong scientific merit. However, in order to be improved, I kindly suggest the authors undergoing the following comments:
1. Page 3, line 105: "estimates" or "retrievals" ? I think the latter is more appropriate.
2. Page 4: In the equation of Transmission, it seems that the integration term is not written with a formula editor and looks like a low-resolution image. Please consider writing throughout the manuscript, all the equations with a professional equation editor.
3. Table 2. I was not able to find in the manuscript what is the full name of the acronym "Nil" used by the authors in this table. Moreover, consider deleting the word "analyzed" appearing for two times in the same sentence.
4. Page 6, line 209: Please define here which are these two wavelenegths. Also, consider specifying if you refer to the entire atmospheric column or only on the volcanic layers?
5. Page 6, line 209: I kindly suggest to the authors to refer to the Ansgtroem exponent with "Å" instead of "α: greek alpha"
6. Page 6, line 217: For the same Icelandic eruption, Kokkalis et al., 2013, found that the range resolved extinction related Angstroem (for the 355 and 532 nm) varies from 0.7 to 1.7. He also demonstrated how the depolarization ratio, lidar ratio and Angstroem exponent of the volcanic layer changes throughout the layers travel from the North down to the Eastern Mediterranean. You are kindly requested to add this information here and in section 4.2, where the results of this study are compared with the relevant literature.
Kokkalis, P., Papayannis, A., Amiridis, V., Mamouri, R. E., Veselovskii, I., Kolgotin, A., Tsaknakis, G., Kristiansen, N. I., Stohl, A., and Mona, L.: Optical, microphysical, mass and geometrical properties of aged volcanic particles observed over Athens, Greece, during the Eyjafjallajökull eruption in April 2010 through synergy of Raman lidar and sunphotometer measurements, Atmos. Chem. Phys., 13, 9303–9320, https://doi.org/10.5194/acp-13-9303-2013, 2013.
7. Page 6, lines 233-234: It is not clear to the reader what is meant by the authors here. They are kindly requested to rephrase.
8. Section 2.4: In my understanding, the authors are using in their analysis the volume linear depolarization ratio (and not the particle). However later on, particle linear depolarization measurements are presented. Please consider introducing in section 2 all the aerosol properties presented in section 3.
One more important thing is how the authors calibrate their depolarization signals. It seems that they are using the molecular atmosphere for this purpose. In any case, they are kindly requested to mention the appropriate reference and to elaborate on the uncertainties introduce my the molecular calibration method followed.
9. Section 2: In my opinion, section 2 needs more work. Since again later on in section 3, various methods (i.e. radiosondes, HYSPLIT) and satellite data (AIRS, OMPS, MODIS, CALIPSO) are presented. The question is: Do they really serve towards the objective of the manuscript?
If yes, the authors should consider at least introduce them in section 2 before presenting them in the Results.
10. Page 7, line 266: "analyses"->"analysis"
11. Page 8, line 283: I wonder why the authors did not use the latest v4.20 data release. Will this change there results?
12. Figure 1: I wonder why the scattering ratio in the lower troposphere is constantly 1. Is this due to the overlap function of the system which constraints its dynamic range in the stratosphere only?
13. Figure 2: I kindly suggest the authors merge this figure with figure 1. They already present many figures and it is beneficiary for the manuscript to merge some of them when and if possible. In this case Figure, 1 and 2 are showing the same property (scattering ratio) before and after the eruption.
14. In my opinion, Figure 6 (a) and (b) should be deleted as long as the authors are presenting the spatiotemporal evolution of the scattering ratio from that site and for both dates (Figures 7 and 8 which by the way can be merged in one figure annotated with (a) and (b))
15. Page 16, lines 478-479: These three methods are giving the same results ? Why the authors choose to apply all the three of them and which one is presented in table 4?
16. Page 16, lines 500 & 504: "Winkler"->"Winker"
17. Table 5: Can the authors briefly explain what is the physical meaning between the total particle linear depolarization and the particle linear depolarization ratios? Why the maximum value is presented and not the mean in this table?
18: Figure 10: "The thins dashed ... ": But this is far away smaller compared to the systematic uncertainties introduced by the system itself and the not proper depolarization calibration method.
19. Section 4.1 This is not a comparison since lidar and radiosondes are providing different datasets. Maybe "Synergy of lidar and radiosonde measurements" is more appropriate
20. Page 25, Lines 818-828: This is a piece of important information but it is not highlighted in the manuscript. This can be done by presenting these results by using figures rather than tables (see table 5 and figures 9 (a) and (b)).
Reviewer 2 Report
The manuscript presents interesting observations of a volcanic ash plume over Australia. The authors follow valid scientific approach, the presentation quality is good (if somewhat long), therefore I suggest that the manuscript should be published. However, some issues should be addressed.
Major comments:
1) In section 2.4, the authors should explain if any calibration procedure is followed to account for different sensitivities in parallel and cross detection modes (e.g. due to different transmissivity of the LCVR at the two states or possible polarization-dependent sensitivity of the detectors). What are the expected errors of the depolarization ratios e.g. due to cross-talk between the two channels?
2) The multiple scattering correction seems wrong. The multiple scattering correction factor should depend on the lidar geometry (FOV and possibly lidar/aerosol distance). I doubt that the factors calculated for a spaceborne lidar like CALIOP are relevant for the systems in this study. See e.g. https://www.osapublishing.org/ao/abstract.cfm?uri=ao-37-12-2464
3) In section 3.4, the authors should clearly define what is the definition of each of the two particle depolarization quantities, and explain what is the analytic relationship between them. Moreover, I suspect that the definition of delta_mol in the two equations (lines 516 and 521) are different, leading to confusion and possibly calculation errors.
Minor notes:
The typesetting of tables is not good, making them hard to read (e.g. in Table 2, the Radiosonde column).
l. 88: Define UTLS when it is first used.
l. 93-94: microphysical parameters would refer to size distribution, refractive index etc. The lidar used in this study are not capable of such measurements. Please rephrase.
Sections 2.1 and 2.2: Please define the detection mode for each lidar system (analog, photon counting, or other)
l. 156: “beta_aer and beta_mo are the molecular and aerosol …”. Please correct the order.
l. 183-184: Give an example of the magnitude of the ozone transmission term up to the detected layer.
l. 195: It is not clear to me why the overlap function would affect the sky background contribution of the measurement.
l. 214: You are using the same symbol for extinction and angstrom exponent; you should change that.
l. 237-239: Add an appropriate image to showcase this procedure. The fit to the molecular signal could give a useful insight about the noise level of the signals.
l. 244: “total backscatter coefficients”. Wouldn’t volume backscatter coefficient be a more consistent terminology?
l. 245: “for co-polarized and cross-polarized”. Please correct the order.
l. 248-249: It is not clear to me what the “mathematical rotation of the phase angle” refers to. Please add a reference to assist the readers understanding this.
l. 332: The feature you are referring to is not very clear.
Figure 6: What is the reason of the apparently different overlap function below 4km in the two panels?
Figure 7: The choice color scale is unusual and probably not very successful. Please note that the scale looks especially bad when printed in black and white, since it is not perceptually uniform.
ll. 458-459: Define z1 and z2
l. 476: Wrong typesetting
Table 4: Which of the three methods was used to calculate these optical depth values?
l. 550 and Table 5: The ash faction refers to ash backscatter fraction (n.b. that the correct reference for the procedure should be this: https://agupubs.onlinelibrary.wiley.com/doi/pdf/10.1029/2009JD011862)
Round 2
Reviewer 1 Report
In the revised version the authors successfully addressed my concerns. Therefore, in my opinion, the manuscript can be published in its current state in the Atmosphere journal of MDPI.